# IDENTIFYING LATENT STATE-TRANSITION PROCESSES FOR INDIVIDUALIZED REINFORCEMENT LEARNING

## ABSTRACT

In recent years, reinforcement learning (RL) has been increasingly applied to systems that interact with individuals in various domains, such as healthcare, education, and e-commerce. When an RL agent interacts with individuals, individual-specific factors, ranging from personal preferences to physiological nuances, may causally influence state transitions, such as health conditions, learning progress, or user selections. Consequently, different individuals may exhibit different state transition processes. Understanding these individualized state-transition processes is crucial for making individualized policies. In practice, however, identifying these state-transition processes is challenging, especially since individual-specific factors often remain latent. In this paper, we present a practical method that effectively learns these processes from observed state-action trajectories, backed by theoretical guarantees. To our knowledge, this is the first work to provide a theoretical guarantee for identifying the state-transition processes involving latent individual-specific factors. Our experiments on synthetic and real-world datasets demonstrate that our method can effectively identify the latent state-transition processes and help learn individualized RL policies. The code is available at CODE.

## 1 INTRODUCTION

Reinforcement Learning (RL) (Konda & Tsitsiklis, 1999) involves training agents to make decisions through interactions with an environment. In this framework, an agent observes its current state, takes an action based on a policy, and receives a reward, which then leads to a new state. This sequence of moving from one state to another is known as the state transition process.

In recent years, RL has been increasingly applied in systems that directly interact with individuals, spanning sectors such as healthcare (Yom-Tov et al., 2017; Liao et al., 2020; Ghosh et al., 2023), education (Shawky & Badawi, 2019; Bassen et al., 2020; Fahad Mon et al., 2023), and e-commerce (Lei & Li, 2019; Yin & Han, 2021; Afsar et al., 2022). When RL systems engage with individuals, they can encounter many individual-specific factors (Lu et al., 2018). These can range from individual preferences and past experiences to physiological differences, all of which can causally influence the transitions between states. For example, in the realm of education, two students with the same background knowledge might progress at different learning rates after watching the same tutorial video, due to their different learning styles. Similarly, in healthcare, two patients diagnosed with hypertension might experience different health outcomes after receiving the same treatment, influenced by factors like genetic predispositions or lifestyle habits.

Understanding individualized state-transition processes is essential for designing better RL systems that can provide more individualized and effective decisions (Kaiser et al., 2019; Hafner et al., 2020; Bennett et al., 2021; Pace et al., 2023). Continuing from the aforementioned example, if an RL system understands a student's learning style, it can recommend resources that better suit their learning styles, such as animated content for visual learners or hands-on exercises for those who learn by doing. In a healthcare context, if an RL system recognizes a patient's genetic makeup, it can suggest treatment plans that align with their specific needs, leading to better health outcomes. This highlights the importance of understanding and identifying individualized state-transition processes.

However, a challenge arises when considering that many of these individual-specific factors are latent and not directly observable. The question of how to learn the latent state transition process which includes these latent individual-specific factors from observed states and actions, remains a

challenging problem. To our knowledge, only a few studies have attempted to uncover the latent state transition process, and none have conclusively determined whether it is possible to identify this process using observable data (Lu et al., 2018; Pace et al., 2023).

In this paper, we propose a novel method that effectively learns these processes from observed state-action trajectories. Our contributions are summarized as follows:

- We propose the Individualized Markov Decision Processes (iMDPs) framework, a novel approach that integrates individual-specific factors into decision-making processes. We model these factors as latent confounders. We allow them to influence each state in the decision process and to vary across different individuals. This framework has many real-world applications.
- Our framework has theoretical guarantees. We show that when individual-specific factors are finite, our method ensures the identifiability of whole latent state-transition processes, even when the transition processes are nonlinear. This establishes novel theoretical insights for learning state-transition processes with latent individual-specific factors. Additionally, for scenarios with non-finite individual-specific factors, we show that categorizing them into finite groups has little impact on the empirical results. To the best of our knowledge, this is the first work to provide a theoretical guarantee for the identification of latent individual-specific factors from observed data.
- We propose a practical generative-based method that can effectively infer latent individual-specific factors. Empirical results on both synthetic and real-world datasets demonstrate the method's effectiveness not only in inferring these factors but also in learning individualized policies.

## 2 RELATED WORK

**Individualized Machine-Learning Applications**   Machine learning in the modern era creates highly individualized solutions across various domains. In health, it customizes interventions for physical activity, weight loss, and diabetes management (Yom-Tov et al., 2017; Liao et al., 2020; Forman et al., 2019; 2023). It assists the elderly with technology and care (Hoey et al., 2014), aids the financial sector in precise stock predictions (Li et al., 2019), and enhances education through personalized ICT systems (Fok et al., 2005; Ji et al., 2017). In transportation, it develops tailored car-following strategies (Song et al., 2023), while in multimedia, platforms like YouTube and TikTok use it for custom video recommendations (Cai et al., 2022; Hoiles et al., 2020). These instances highlight machine learning's broad applicability in today's world.

**Reinforcement Learning for Latent State-Transition Processes**   RL has made significant progress, especially with latent variable models for understanding environment dynamics. The main aim is learning low-dimensional, latent Markovian representations from data (Lesort et al., 2018; Krishnan et al., 2015; Karl et al., 2016; Ha & Schmidhuber, 2018; Watter et al., 2015; Zhang et al., 2018; Kulkarni et al., 2016; Mahadevan & Maggioni, 2007; Gelada et al., 2019; Gregor et al., 2018; Ghosh et al., 2019; Zhang et al., 2021). Strategies include reconstructing observations, learning forward or inverse models, and using prior knowledge like temporal continuity (Wiskott & Sejnowski, 2002). Various studies (Watter et al., 2015; Ebert et al., 2017; Ha & Schmidhuber, 2018; Hafner et al., 2018; Zhang et al., 2019; Gelada et al., 2019; Kaiser et al., 2019; Hafner et al., 2020) propose methods for estimating state-transitions from high-dimensional sequences, aiding model-based RL or planning. Recent research (Lu et al., 2018; Li et al., 2020; Vo et al., 2021; Wang et al., 2021; Bennett et al., 2021; Pace et al., 2023) focuses on estimating state-transitions with latent confounders. Some works (Lu et al., 2018; Pace et al., 2023) address similar scenarios with individual-specific factors, but a systemic approach for clear identifiability in such settings is still lacking.

## 3 PRELIMINARIES

To model the individualized property given population samples, we propose Individualized Markov Decision Processes (iMDPs), which is an extension of the standard MDP with latent confounders. We assume that the latent confounders have a long-term and constant influence on the entire decision-making process, while the factors differ from individual to individual.

**Definition 3.1.** An Individualized Markov Decision Process (iMDP) is a tuple $\langle \mathcal{S}, \mathcal{A}, \mathbb{P}_l, \mathcal{R}, \gamma \rangle$, where $\mathcal{S}$, $\mathcal{A}$, and $\mathcal{R}$ are the state, action, reward spaces, respectively, and $\gamma$ is the discount factor. $\vec{s}_t = (s_{0t}, \ldots, s_{nt})^\top \in \mathbb{R}^n$ and $\vec{a}_t = (a_{0t}, \ldots, a_{mt})^\top \in \mathbb{R}^m$ represent the state and action vectors at time $t$, respectively. $r_t = R(\vec{s}_{t-1}, \vec{a}_{t-1}, \vec{s}_t) \in \mathbb{R}$ denotes the immediate reward. $l \in \mathcal{L}$ are the

specific latent individual-specific factors where $\mathcal{L}$ is the latent individualized space. We define the corresponding individualized state transition distribution $\mathbb{P}_l$ as

$$\mathbb{P}_l(\vec{s}_t|\vec{s}_{t-1}, \vec{a}_{t-1}) = \mathbb{P}(\vec{s}_t|\vec{s}_{t-1}, \vec{a}_{t-1}, l), \tag{1}$$

which indicates that the dynamic transition from the history $(\vec{s}_{t-1}, \vec{a}_{t-1})$ to the current state $\vec{s}_t$ is influenced by the latent individual-specific factors $l$.

In this context, the population samples can be viewed as a family of iMDPs $\{\langle \mathcal{S}, \mathcal{A}, \mathbb{P}_l, \mathcal{R}, \gamma \rangle | l \in \mathcal{L}\}$. Different from the majority of latent MDP research which focuses on the time-varying latent (Feng et al., 2022; Zhang et al., 2020; Guo et al., 2020), in our work, the latent confounders are assumed to influence each state in the decision process. It remains constant for each individual, representing the unique characteristics, and varies between different individuals, highlighting the individual differences among them. A graphical representation of the data generation process is shown in Figure. 1(a).

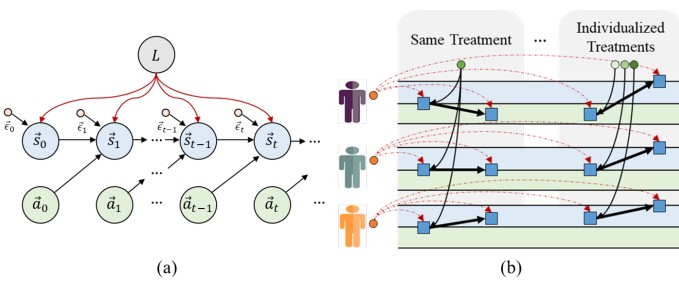

This formulation is intuitive in many real-world applications. For example, in the context of hypertension diagnosis as shown in Figure. 1(b). The patient's health status at each time can be represented as the state $\vec{s}_t$, the individualized treatment can be viewed as the action $\vec{a}_t$, and the blood pressure level is the reward $r_t$. The latent confounder $l$ represents the unobserved genetic predispositions that are unique to each patient and have a significant

Figure 1: (a) Data generation processes. (b) Hypertension diagnosis.

cant impact on their long-term health, such as gene variations. In this case, the health status $\vec{s}_t$ at time $t$ is determined by the previous status $\vec{s}_{t-1}$, previous treatment $\vec{a}_{t-1}$, together with the intrinsic gene variations. Such implicit associations are indicated by the red dashed lines. If we could observe these predispositions, it would greatly help physicians to predict how patients will respond to specific treatments, enabling personalized care. In most cases, however, these individualized factors remain latent and difficult to diagnose. Therefore, the theoretical identification of such traits and the development of an estimation framework to extract them from observed data is of great practical value.

**Data Generation Process**  Consider a population with $K$ unique individuals that can be divided into $q$ groups, where the exact group membership is unknown. A trajectory for individual $k$ $(k = 1, \ldots, K)$ is a sequence of state-action tuples experienced by an agent during its interaction with the environment, and is denoted as $\tau_k = \{(\vec{s}_0^k, \vec{a}_0^k), (\vec{s}_1^k, \vec{a}_1^k), \ldots, (\vec{s}_T^k, \vec{a}_T^k)\}$ with $T$ being the total length of the trajectory. It is important to note that these samples are assumed to be generated from a stationary process for each individual, but show heterogeneity across different groups.

Here we introduce the individualized transition processes. Suppose that the observed states of the $k$-th individual $\vec{s}_t^k = (s_{0t}^k, \ldots, s_{nt}^k)^\top$ satisfy the following generation process

$$s_{it}^k = f_i(\vec{s}_{t-1}^k, \vec{a}_{t-1}^k, l^k, \epsilon_{it}^k), \tag{2}$$

for $i = 1, \ldots, n$. Here $\epsilon_{it}^k$ is the noise term, and $l^k \in L$ is the latent individual-specific factor. From the population perspective, $L = \{l_1, l_2, \ldots, l_q\}$ is the collection of all latent confounders with $q$ being the cardinality of $L$. Furthermore, $L$ characterizes the unique properties of the transition function $f$ across different individuals, which is consistent with Equation 1.

**Problem Setting**  In this work, we focus on developing individualized policies within RL. We aim to (1) identify the latent individual-specific factors $L$ from the observed sequential trajectories, and (2) develop individualized policy learning as well as policy adaptation for new individuals. Consider the hypertension diagnosis example in Figure. 1(b). The same treatment applied to different individuals would yield different results because the state transitions dynamics are influenced by the latent confounders. Therefore, recovering $L$ from the population data is important for accurate dynamic prediction, which is consistent with our first goal. After identifying $L$, we can then divide the population into different subgroups and make individualized treatment for each patient to achieve the expected outcomes, aligning with our second goal.

# 4 IDENTIFIABILITY ANALYSIS

Following prior work, we define identifiability in representation function space.

**Definition 4.1** (Identifiability). Let $\vec{s}_t$ be a sequence of observed variables generated by the true individualized transition processes specified by $(f_i, l, p_{\epsilon_i})$ given in the preliminary. A learned generative model $(\hat{f}_i, \hat{l}, \hat{p}_{\epsilon_i})$ is observationally equivalent to $(f_i, l, p_{\epsilon_i})$ if the joint distribution $p_{\hat{f}, \hat{l}, \hat{p}_\epsilon}(\vec{s}_t)$ matches $p_{f,l,p_\epsilon}(\vec{s}_t)$ everywhere. We say the latent individualized-specific factors are identifiable if observational equivalence can always lead to identifiability of the latent variables:

$$p_{\hat{f}, \hat{l}, \hat{p}_\epsilon}(\vec{s}_t) = p_{f,l,p_\epsilon}(\vec{s}_t) \Rightarrow \hat{l} = l. \tag{3}$$

Theorem 1 shows the identifiability of the latent individual-specific factors in the individualized transition model. A more detailed explanation of the assumptions and the full proof can be found in Appendix A. Suppose the sampled individuals are from different unobserved groups, and the transition dynamics are different across groups but identical within each group.

**Theorem 1.** *Assume the individualized transition processes in Eq. 2, where the nonlinear state transition functions $f_i$ are stationary within each individual but exhibit variability across different individuals. The different values of $l_k \in L$ describe different transition processes determined by $f_i^k$ for each individual. Here we assume:*

- *(Group Determinacy): The individual-specific factors $l_k \in L$ delineates distinct groups within the finite mixture model, each of which defines and dictates the individualized transition dynamics $f_i^k$.*

- *(Sample Sufficiency): The length of the sequence for each individual $l_s$ is greater than $2q - 1$, where q is the number of groups.*

- *(Sequential Dependency): At any time t, $\vec{s}_t$ is determined only by the given conditions $(\vec{s}_{t-1}, \vec{a}_{t-1}, L)$ within each group, and there are no instantaneous relations between $\vec{s}_t$.*

*Then the identifiability property of the latent individual-specific factors $L$ is ensured.*

**Intuitive Explanation on Assumptions** Group determinacy indicates that for a given task, the individual-specific factors are finite. This is a realistic assumption in numerous practical situations. For instance in hypertension diagnosed, even if they receive the same treatment they might experience different health outcomes. These differences can be attributed to latent factors like their medical history, which are essentially finite records of past medical events and treatments. In this case, we can group patients based on their history. Each group would demonstrate unique health outcomes and patterns, driven by their respective individual-specific factors $l_k$.

The sample sufficiency assumption provides a minimum number of observed samples for reliable identifiability. Such a threshold guarantees enough data diversity to distinguish group characteristics. The sequential dependency assumption further ensures Markov behavior in the dynamic evolution, where only the previous information, such as state and action, influences the current state.

Note that even the required assumptions sometimes may not be held in practice, our method can still encourage the identifiability in different cases, that is: (1) multiple independent latent confounders with cardinalities $c_1, \ldots, c_m$, (2) multiple independent continuous latent confounders, and (3) insufficient samples. In the first case, we can treat multiple latents as one discrete latent with cardinality $\prod_{i=1}^m c_i$. Then each combination of the original multiple latents can be uniquely represented as one level in the combined latent variable. In the second case, the empirical results in Section 6 show that our proposed framework can also encourage the identifiability of the continuous latents. This is because we can divide the continuous latent confounder $L$ into small segments and represent each with a discrete value $l_k$. This discretization approach improves computational simplicity and stability. In the third case, even when the sample sufficiency assumption is weakly violated, our framework still encourage the identification of the latent confounder, as verified in the experiment in Section 6.

**Proof Sketch** According to the group determinacy assumption, each unique instance of $l_k$ represents a distinct group within the mixture model. Therefore, each component of the mixture model is associated with a unique transition function $f_i$. Since there are no instantaneous relations within $\vec{s}_t$, the observed $\vec{s}_t$ generated with $l_k$ are considered to be drawn from the same group of the mixture

model. Thus for each group represented by $l_k$, we have a unique process generating the observations, and the total observations are the mixture of the observations from each group. Lemma 1 in the Appendix states that the identifiability of mixture models from grouped samples would be ensured if the observations are sufficient. According to the sample sufficiency assumption, since we have at least $2q - 1$ sequential observation tuples in each individual, we can uniquely identify any mixture of $q$ probability measures. To summarize, in the individualized transition processes, since the groups are indicated by distinct $l_k$, if the number of samples is sufficient, then the latent confounder $L$ representing the unique group from which the observed samples are drawn can be identified.

## 5 ESTIMATION AND POLICY LEARNING FRAMEWORK

In this section, we propose Individualized Policy learning through Latent Factors estimation (IPLF), a two-stage learning method for optimizing individualized policies. IPLF uses a collection of individual trajectories stored in a buffer, which are meticulously analyzed to 1) construct an estimation framework to recover the individual-specific factors, and 2) establish the individualized policy learning framework to realize policy adaptation for new individuals. The visualization of the learning process is shown in Figure. 2, which aligns with the identifiability theorem of $L$ provided in Section 4. The pseudocode of the entire framework is available in Appendix C.

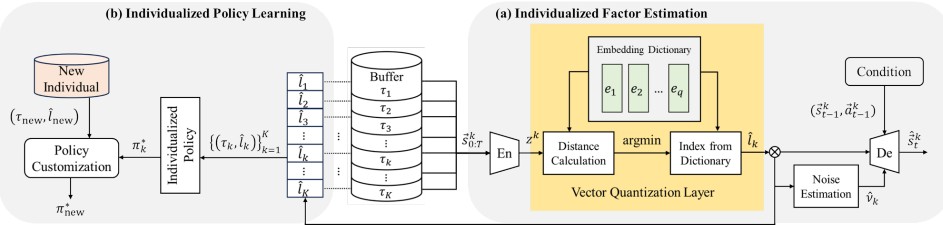

Figure 2: For each individual $k$, (a) Individual-specific Factors Estimation runs the sequence $\vec{s}_{0:T}^k$ through a quantized encoder to estimate the latent confounder $\hat{l}_k$. Then the noise $\hat{\nu}_k$ is estimated to compensate for the determinacy of the quantization operation. Finally, the conditional decoder is designed to mimic the individualized transition processes with $(\vec{s}_{t-1}^k, \vec{a}_{t-1}^k)$, $\hat{l}_k$ and $\hat{\nu}_k$ as inputs, $\hat{\vec{s}}_t^k$ as output. (b) Individualized Policy Learning augment the original buffer with the corresponding latent factors. Then the augmented dataset is processed in a model-based RL fashion to optimize individualized policies and efficiently adapt to new individuals.

**Overview** Our proposed framework is carefully customized to meet the requirements of the identifiability theorem. According to the identifiability definition, the latent confounders are identifiable if observational equivalence can always lead to identifiability of the latent variables. This motivates our work to use a generative model and to realize confounder estimation by fitting the learned distribution to the true observed distribution. The traditional VAE (Kingma & Welling, 2013) encodes the input data into a continuous latent space using probabilistic encoders and then reconstructs the input from this space using decoders, which is not aligned with our problem setting. Specifically, given the offline trajectory of the population, we customize the VAE framework and develop an individual-specific factor estimation framework to recover the latent individual-specific factors.

First, in order to obtain latent representations in a reduced dimensional space, an encoder is used with the entire sequence $\vec{s}_{0:T}^k$ of each individual $k$ as input and the latent continuous representation $z^k$ as output. Considering the group determinacy assumption, the estimated latent confounder should be discrete. We then embed a quantization layer to discretize the continuous latent representation $z^k$ to the latent confounder $\hat{l}_k$. Intuitively, the better $\hat{l}_k$ is recovered, the better the data generation process could be learned. To reconstruct individualized transition processes, we design a conditional decoder that takes the conditions $(\vec{s}_{t-1}^k, \vec{a}_{t-1}^k)$, the estimated confounder $\hat{l}_k$, and the noise term $\hat{\nu}_k$ as inputs, with $\hat{\vec{s}}_t^k$ as the output. By introducing noise, we model the stochasticity in our defined transition process. After that, the optimized quantized encoder is used to augment the original buffer and facilitate the downstream individualized policy learning.

### 5.1 INDIVIDUAL-SPECIFIC FACTOR ESTIMATION FRAMEWORK

**Inferring Individual-Specific Factors via a Quantized Encoder** The group determinacy assumption implies the existence of the latent individual-specific factor $L$. Since $L$ denotes the time-invariant latent individual-specific factors that influence each state in the transition process, we initially use an

encoder to capture the continuous representation $z^k$ from $\vec{s}_{0:T}^k$, and employ a quantization layer to delineate unique and distinct groups as $\hat{l}_k$.

To capture the long-term temporal dependence between the observations, sequential neural networks are used to extract the low-dimensional representation $z^k = g(\vec{s}_0^k, \ldots, \vec{s}_T^k)$ from raw sequences with $g$ being the encoder function. Specifically, we use Conv1D (LeCun et al., 1989) for the synthetic dataset and LSTM (Hochreiter & Schmidhuber, 1997) for the corpus dataset. In the Conv1D, a series of 1d convolutional layers process each subsequence $\vec{s}_{t:t+H}^k$ and traverse the entire sequence to capture local temporal patterns. In the LSTM, each $\vec{s}_t^k$ is processed sequentially, incrementally accumulating information in the hidden states $h_t$ at each time step. The corresponding update functions are

$$o_t^k = \text{Conv1D}(\vec{s}_{t:t+H}^k), \quad h_t^k, c_t^k = \text{LSTM}(h_{t-1}^k, c_{t-1}^k, \vec{s}_t^k; \theta), \tag{4}$$

where $o_t^k$, $h_t^k$ and $c_t^k$ are the feature map, hidden state and cell state at time $t$, and $\theta$ denotes all the parameters of the LSTM. After the sequential process over the entire trajectory, the final hidden state of the LSTM as well as the final output of the convolutional layer serve as the low-dimensional continuous representation $z^k$. Please refer to Appendix F for more details.

Since the derived $z^k$ yields continuous latent representations, a vector quantization (VQ) layer (Van Den Oord et al., 2017) is then employed to delineate distinct groups. This layer maps each continuous representation to the nearest vector in a predefined embedding dictionary $E$ and translates the continuous representations into discrete latent space. Specifically, the embedding dictionary consists of a set of vectors $E = \{e_1, e_2, \ldots, e_q\}$, each of which symbolizes a distinct group in the discrete embedding space. The assignment of a dictionary vector $e_i$ to $z^k$ can be realized by finding the nearest neighbor in the dictionary: $\hat{l}_k = \arg\min_{e_i} \|z^k - e_i\|_2$, where $\hat{l}_k$ represents the quantized vector that is the closest embedding $e_i$ to the continuous representation $z^k$. This component better aligns the representation learning process with our assumptions about the group nature of the latent, making it a reasonable choice for our purposes.

**Estimating Noise for Accommodating Stochasticity of State-Transition Processes**  To account for the stochastic nature of the individualized transition processes, a noise estimator is introduced with $\hat{l}_k$ as input and $\hat{\nu}_k$ as output. Empirically, this noise is modeled using an MLP (Multilayer Perceptron) layer. Then, the output of the noise layer is used as the input of the conditional decoder.

The motivations for introducing a noise layer are 1) to mimic the probabilistic property of the individualized state transition in Equation 1, and 2) to compensate for the loss of stochasticity due to the deterministic nature of the quantization operation. While standard VAEs use the sampling process in the latent space to introduce variability, the categorical and deterministic nature of our posterior reduces this stochastic element. In contrast to standard VAEs, which optimize a variational lower bound on the log-likelihood with the KL divergence penalizing deviation from a given prior distribution, our posterior $q(\hat{l}_k = \kappa | \vec{s}_{0:T}^k) = \begin{cases} 1 & \text{for } \kappa = \arg\min_j \|z^k - e_j\|_2 \\ 0 & \text{otherwise} \end{cases}$ is a categorical distribution and becomes deterministic after the quantization operation. By adding noise to the decoder, we introduce variability that 1) satisfies our individualized transition processes, and 2) allows the model to better vary the data. The empirical results of the ablation study validate the model's enhanced ability to improve generalization.

**Learning State-Transition Processes via Action-State Reconstruction**  To reconstruct individualized transition processes, a conditional decoder is built, which aligns with the sequential dependency assumption. It uses the previous state and action tuples $(\vec{s}_{t-1}^k, \vec{a}_{t-1}^k)$ as conditions to guide the reconstruction on $\hat{\vec{s}}_t^k$. These conditions, together with the estimated latent individual-specific factors $\hat{l}_k$ and the noise term $\hat{\nu}_k$, serve as the inputs to the decoder. The accuracy of this reconstructed state is quantitatively evaluated by its reconstruction likelihood $p_{\text{Recon}}(\vec{s}_t^k | \vec{s}_{t-1}^k, \vec{a}_{t-1}^k, \hat{l}_k, \hat{\nu}_k)$, where $p_{\text{Recon}}$ is the decoder distribution, offering a probabilistic measure of how accurately $\hat{\vec{s}}_t^k$ reconstructs $\vec{s}_t^k$, and providing a quantitative evaluation of the model's reconstruction accuracy.

**Training Objectives**  During the training process, the parameters are optimized according to the extended ELBO objective $\mathcal{L}_{\text{ELBO}}$.

$$\mathcal{L}_{\text{ELBO}} = \mathcal{L}_{\text{Recon}} + \beta \mathcal{L}_{\text{Quant}} + \alpha \mathcal{L}_{\text{Commit}} \tag{5}$$

where $\alpha$ and $\beta$ are weights for the corresponding loss components. Specifically, (1) the reconstruction loss $\mathcal{L}_{\text{Recon}} = \sum_t \|\vec{s}_t^k - \text{De}(\text{En}(\vec{s}_{0:T}^k), \vec{s}_{t-1}^k, \vec{a}_{t-1}^k, \hat{\nu}_k)\|^2$ measures the discrepancy between the reconstructed state $\hat{\vec{s}}_t^k$ and the original state $\vec{s}_t^k$. Here En and De denote the encoder and decoder, respectively. (2) The quantization loss updates the embedding space by evaluating the difference between the encoder output $z^k$ and the discretized representation on $e^k$ obtained by vector quantization, defined as $\mathcal{L}_{\text{Quant}} = \sum_i \|\text{sg}[z_i^k] - e_i^k\|^2$, where sg is the stop-gradient operator. (3) Commitment loss ensures that the encoder's outputs are consistent with the embedding space, thereby decrease variations in the encoder's output (e.g., switching from one embedding vector to another). This regularization term is computed as the L2 error between $z^k$ and $e^k$, represented as $\mathcal{L}_{\text{Commit}} = \sum_i \|e_i^k - \text{sg}[z_i^k]\|^2$.

## 5.2 INDIVIDUALIZED POLICY LEARNING FRAMEWORK

**Leveraging Individual-Specific Factors For Individualized Policy**   The estimated individual-specific factors $\hat{l}_k$, along with the trajectories stored in the buffer, are used to adjust the individualized policy $\pi_k^*$. Our framework is general enough to be integrated with many RL algorithms. In general, the conventional policy input is extended to include latent factors specific to each individual, and the training objective is adjusted accordingly to prioritize the optimization of policies tailored to the individual. Here we use Deep Deterministic Policy Gradient (DDPG) (Lillicrap et al., 2015) as an example. Traditional DDPG learns an optimal policy by maximizing the expected return for continuous action spaces. In our individualized approach, the input includes the latent factors, and the policy is updated as $\mu_\pi(\vec{s}_t; \theta^\mu) \rightarrow \mu_\pi^K(\vec{s}_t^k, \hat{l}_k; \theta^\mu)$, where $\theta^\mu$ denotes the parameters of the policy network. Incorporating latent factors into a DDPG allows the model to better tailor policy to the unique characteristics of different individuals. The training objective is then updated as $J(\theta^\mu) = \mathbb{E}\left[\sum_{t=0}^{\infty} \gamma^t Q\left(\vec{s}_t, \mu_\pi^K(\vec{s}_t^k, \hat{l}_k; \theta^\mu); \theta^Q\right)\right]$, where $\gamma$ is the discount factor and $Q$ is the Q value. The individualized DDPG develops customized policies for each individual, increasing adaptability in changing environments.

**Policy Adaptation for New Individual**   After optimizing the individualized policy, the agent uses $\pi_k^*$ as a warm start, which is an initial starting point that helps speed up training, and continues training on the new individual. To achieve zero-shot transfer, we simultaneously optimize the individualized policy while interacting with the environment. Thus, adaptation involves a dual process: it fine-tunes the policy based on interactions with the new individual, and it actively collects data from these interactions. This data collection is critical for estimating the latent individual-specific factor of the new individual. During this phase, the policy is continuously adjusted and improved, making it better suited to each new individual.

## 6 EXPERIMENT

We comparatively evaluate the proposed method on a number of temporal datasets to verify the performance of (1) individual-specific factor estimation and (2) individualized policy learning.

**Evaluation Metrics**   To measure the identifiability of the latent individual-specific factors, we use (1) Pearson Correlation Coefficient (PCC) for singular latent case and (2) Canonical Correlation Analysis (CCA) for multiple latents case, implementing these on the test dataset to quantify the correlation between the estimated and actual latent variables. A value nearing 1 denotes a strong correlation between the variables, whereas a value approaching 0 indicates a weak correlation. To evaluate the control performance, we consider (3) jumpstart and (4) accumulative reward. Jumpstart refers to the improvement in the initial performance when a learning agent leverages knowledge transferred from a source task. Accumulative reward indicates the learning quality and transfer success, typically measured by the rewards earned as the model adapts to a new environment, offering valuable insight into the learning quality.

**Baselines**   (1) Disentangled Sequential Autoencoder (DSA) (Yingzhen & Mandt, 2018), which separates the latent representations into static and dynamic parts instead of considering the global influence of $L$. (2) Population-Level Component (PLC), which learns the population-level embeddings instead of focusing on individualized factors. (3) VQ-VAE (Van Den Oord et al., 2017) serves as the base model for ablation. Model variants are built upon it to disentangle the contributions of different modules. (4) Adaptation to continuous latent variables, which allows us to evaluate the flexibility of our method and to assess if the continuous latent assumption distorts identifiability.

6.1 EVALUATION ON ESTIMATION FRAMEWORK

**Synthetic Experiments** We manually created time series data based on the post-nonlinear model (Zhang & Hyvarinen, 2012) and customize different types of latents with the required assumptions satisfied or violated. The data generation process is modeled as $\vec{s}_t = f_2(f_1(\vec{s}_{t-1}, \vec{a}_{t-1}, L), \epsilon_t)$, where $f_1$ represents the nonlinear effect, and $f_2$ denotes the invertible post-nonlinear distortion on $\vec{s}_t$. The state $\vec{s}_t \in \mathbb{R}^3$ and actions $\vec{a}_t \in \mathbb{R}^2$ are initially generated randomly following a uniform distribution, and the noise is modeled as a Gaussian distribution. Three types of $L$ are generated to satisfied or violate our assumptions. Case 1: $L$ is a discrete latent scalar follows the categorical distribution $Cat(0.1, 0.2, 0.3, 0.4)$. Case 2: $L$ are multiple discrete latent variables follow the categorical distributions $Cat(0.1, 0.9)$, $Cat(0.2, 0.3, 0.5)$, $Cat(0.1, 0.2, 0.3, 0.4)$ respectively. Case 3: $L$ are multiple continuous latent variables follow the Gaussian distribution $\mathcal{N}(0, 1)$, uniform distribution $Uniform(0, 1)$, and exponential distribution $Exp(1)$. During the data collection period, we generate 100 individuals, with the maximum total length of each trajectory being 300.

Fig. 3(a) and Fig. 3(b) present results with Case 1. The comparison result in Fig. 3(a) shows that our method outperforms the baselines. Specifically, DSA reconstructs the entire time series but overlooks individualized transition processes, resulting in compromised identifiability and deteriorating recovery performance over time. PCL lacks the individual-specific factor to delineate distinct groups thus cannot achieve identifiability. Fig. 3(b) shows the relationship between sequence length and the identifiability performance. It shows that satisfying sample sufficiency assumption is necessary to recover the latent variable, while usually, the longer the length, the better the performance. Fig. 3(c) and Fig. 3(d) illustrate the identifiability results in Case 2 and Case 3. The high CCA values indicate the successful recovery of latent variables, even when our assumptions are violated. This suggests the potential applicability of our model in guiding the learning of multiple latent individual-specific variables, reinforcing its versatility and practical relevance.

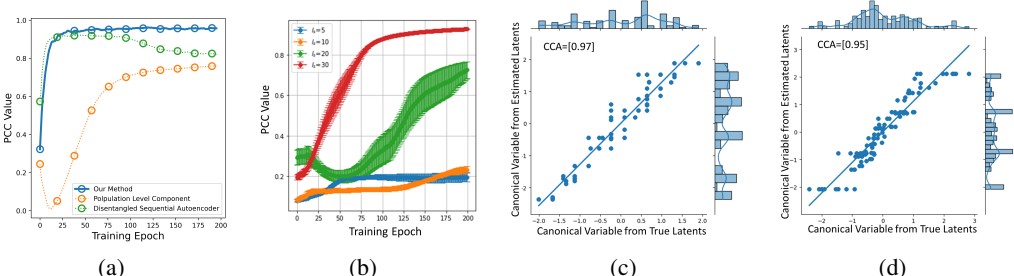

(a)       (b)       (c)       (d)

Figure 3: Synthetic experiment results. (a) PCC trajectory comparisons with baselines. (b) Identifibility performance responses of individual sequential length. Values are means$\pm$SD, $n = 5$. (c) Scatterplot of canonical variables in Case 2. (d) Scatterplot of canonical variables in Case 3.

**Ablation Study** We show the contributions of different components in the Individual-specific Factor Estimation in Table 1. The conditional decoder primarily contributes to a reduction in model bias, illustrating its usefulness in refining the accuracy of the model. The addition of a sequential encoder, using Conv1D as an example, can significantly im-

Table 1: Contribution of each module.

| Module | PCC | Bias |
|---|---|---|
| VQ-VAE | $0.646 \pm 3.1e\text{-}04$ | $0.099 \pm 2.3e\text{-}04$ |
| + Conditional Decoder | $0.382 \pm 3.8e\text{-}02$ | $0.081 \pm 2.8e\text{-}06$ |
| + Conv1d Encoder | $0.910 \pm 1.3e\text{-}04$ | $0.077 \pm 5.7e\text{-}06$ |
| + Noise Estimator | $0.942 \pm 4.0e\text{-}05$ | $0.072 \pm 3.0e\text{-}07$ |

prove the identifiability of the model, highlighting its importance for accurate latent variable recovery. Finally, the use of a noise estimator further optimizes the model, reducing bias while improving identifiability. The result implies that the noise estimator is instrumental in fine-tuning the overall performance of the model, allowing for more accurate and reliable extraction of latent confounder.

**Persuasion For Good Corpus** We further evaluate the effectiveness of the estimation framework on a real-world, open-source dataset, the Persuasion For Good Corpus (Zhang et al., 2017; Wang et al., 2019). It is a collection of dialogues focused on charitable donations, and each dialog involves a persuader trying to convince a persuadee to donate to a charity, where all participants have undergone personality assessments and have corresponding individualized personalities. The state is the persuadee's response, the action is the persuader's utterance, and the reward is the final donation in each dialog. The goal is to make an individualized policy for each persuader to successfully

persuade the persuadee to make a charitable donation. To achieve this, we first need to identify the latent personality of each individual from the offline dataset.

We initially employ a pre-trained BERT model (Devlin et al., 2018) to convert natural language (dialogues) into vectors, then use an LSTM encoder to extract latent personalities. Fig. 4(a) shows the curve of the first canonical correlation relative to the latent dimension, demonstrating that by carefully selecting the latent dimensions in real data, our method can achieve commendable performance.

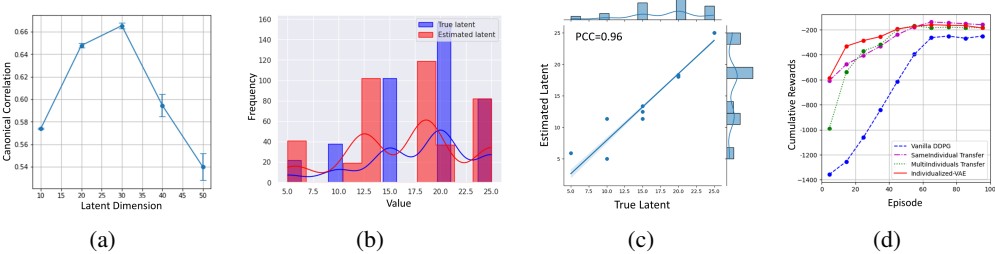

(a)   (b)   (c)   (d)

Figure 4: In corpus dataset: (a) The first canonical correlation relative to the latent dimension. In Pendulum dataset: (b) Frequency comparisons of the true latent with the estimated latent. (c) Scatterplot of the true latent with the estimated latent. (d) Training reward curve compared with baselines.

## 6.2 EVALUATION ON POLICY LEARNING FRAMEWORK

The pendulum environment from OpenAI Gym is a classic control task for RL studies. It presents a continuous control task, where the agent aims to control a frictionless pendulum, swing and stabilize it in its inverted position with minimal effort. For a pendulum of length $l$, mass $m$, gravity $g$, and a continuous control input $u$, the dynamics is $ml^2\ddot{\theta} + mgl\sin(\theta) = u$, where $\theta$ is the angle of the pendulum and $\ddot{\theta}$ is the angular acceleration. We choose DDPG as the RL algorithm and manually create 20 individualized environments, each changing the gravity as $\{5.0, 10.0, 15.0, 20.0, 25.0\}$ and $g$ follows a categorical distribution $g \sim \text{Cat}(0.1, 0.2, 0.1, 0.4, 0.2)$. A buffer is populated with 100 time steps for each individual, and the new individual is assumed to have gravity $g = 10.0$.

Fig. 4(b) and Fig. 4(c) show the successful recovery of the latent individual-specific factor, validated by high-frequency similarity and a remarkable PCC value, confirming the ability of our method to skillfully recover latent variables in practical RL tasks. For the individualized policy learning, we compare our method with (1) standard DDPG with no prior training, (2) a pre-trained policy network using data collected from the new individual—termed SameIndividual Transfer, and (3) a pre-trained policy network using data collected from different unknown grouped individuals - termed MultiIndividual Transfer. The comparative results are shown in Fig. 4(d). The performance trends of all transferred policies illustrate that incorporating a pretrained policy model accelerates the learning process, a benefit attributed to jumpstart and accumulative reward. However, in the absence of individual-specific information, as in the case of MultiIndividual Transfer, the mixed policy training yields inferior initial performance compared to the individualized policies derived from IPLF , as evidenced by a jumpstart comparison of -600 versus -1000. Interestingly, the use of individualized training not only introduces robustness but also allows IPLF to converge faster than SameIndividual Transfer, reflecting the advantage of precise and individual-centric learning policies in promoting accelerated model convergence.

## 7 CONCLUSION AND FUTURE WORK

Our paper presents a method to learn latent state transitions from the observed state-action trajectories, ensuring the identifiability in the presence of latent individual-specific factors. Empirically, the effectiveness of our method is validated in inferring the latent confounder and in learning individualized policies. The main limitations of this work lie in our two main assumptions: (1) there is no instantaneous causal influence within $\vec{s}_t$, and (2) the latent confounder is discrete. Instantaneous relationships, if present, may distort the identifiability of the latent state-transition process, although their impact can be moderated by adjusting the temporal resolution of the data. Although our theoretical framework does not extend to scenarios with evolving continuous latents, our empirical results suggest potential for adapting our approach to a broader range of scenarios. Extending our identifiability theories and framework to account for such properties is our future direction.

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
