# Supplementary Materials for "Identifying Latent State-Transition Processes for Individualized Reinforcement Learning"

## A  IDENTIFIABILITY THEORY

### A.1  NOTATION AND TERMINOLOGY

We summarize the notations used throughout the paper in the following table.

| **Index** | |
| --- | --- |
| $t$ | Time index |
| $\tau$ | Trajectory |
| $K$ | Number of individuals |
| $k$ | Index for a specific individual |
| $q$ | Number of groups |
| $T$ | Total length of time series |
| $i, j$ | Variable element index |
| $\beta, \alpha$ | Weights in the augmented ELBO objective |
| **Variable** | |
| $\vec{s} = [s_1, s_1, \ldots, s_n]^\top$ | $n$-dimensional observed states |
| $\vec{a} = [a_1, a_2, \ldots, s_m]^\top$ | $m$-dimensional observed actions |
| $L$ | Latent individual-specific factor |
| $l_k$ | The value of $L$ for a specific individual $k$ |
| $\hat{\vec{s}}_t$ | Reconstructed states at time $t$ |
| $\epsilon_{it}$ | Process noise term |
| $\hat{\epsilon}_{it}$ | Estimated process noise term |
| $l_s$ | Length of the sequence for each individual |
| $f_i$ | Nonlinear transition function for $s_{it}$ |

### A.2  DISCUSSION ON ASSUMPTIONS

Here, we provide intuitive explanations for each assumption and discuss their relevance to real-world applications.

#### A.2.1  GROUP DETERMINACY ASSUMPTION

The value $l_k$ in $L$ is key for the transition function $f$, as it specifies how $f$ behaves differently for each person. Essentially, $L$ categorizes the population into groups, each with its own probability distribution and transition behavior represented by $f$. In this model, each individual is part of a unique group, each with its own transition dynamics defined by a specific function $f^k$.

**Applicability**   The idea of group determinacy is important in real-world applications, such as diagnosing hypertension in healthcare. Here, $\vec{s}^k$ represents the health condition of a patient, $\vec{a}^k$ is their tailored treatment, and the function $f^k$ decides the unique health trajectory for each patient group. The latent individual-specific factor $l_k$ influences how a patient's health changes over time. It can be based on factors such as age, gender, or medical history that help to group patients logically. For example, one group might be younger people without hypertension, and another group might be older people with different medical histories. Each group has its own set of health outcomes and patterns, influenced by $l_k$ and guided by $f^k$.

#### A.2.2  SAMPLE SUFFICIENCY ASSUMPTION

In a finite mixture model with $q$ groups, each group requires sufficient observations to identify the latent individual-specific factor $L$. This assumption provides a minimum number of observation samples, requiring more than twice the number of groups minus one $(2q - 1)$ observations. This

threshold ensures that we have enough information and variability in the observed data to distinguish the characteristics of each group. Essentially, we should have a variety of transition samples for each group to ensure that we can correctly identify each group in the model. This assumption helps to identify the unique characteristic of each individual, which is critical for identifibility.

**Applicability**    Sample Sufficiency indicates that sufficient data is needed to achieve identifiability, which should be a common assumption. For example, in nonlinear ICA using auxiliary variables (Hyvarinen et al., 2019), it assumes that there should be at least 2n+1 values for auxiliary variables to encourage variability; for disentanglement with minimal change (Kong et al., 2023), it assumes that there should be at least 2n+1 domain embeddings to ensure identifiability. Intuitively, without having data to provide us with related information about parameters, it is impossible to "determine" the value of these parameters.

### A.2.3   SEQUENTIAL DEPENDENCY ASSUMPTION

At any given time $t$, the observations $\vec{s}_t$ depend solely on the specific conditions within each group. This assumption implies that the elements of $\vec{s}_t$ have no instantaneous connections with each other, and that the state and action at time $t-1$ along with $L$ are the only influences on the state at time $t$. For each element $i$ at time $t$, the previous state and action $(s_{it-1}, a_{it-1})$ is independent of $L$. In other words, each element of $\vec{s}_t$ is independently influenced by its previous state $\vec{s}_{t-1}$, its previous action $\vec{a}_{t-1}$, and the latent individual-specific factor $L$.

**Applicability**    Sequential dependency is commonly seen in other related works, such as Temporal Causal Mechanisms in Block MDP (Zhang et al., 2020). It ensures that only previous information (such as state and action) influences the current state. In the case of individualized healthcare, this assumption implies that a patient's latent factors (such as medical history), current health state and current treatment decisions determine their next health state.

### A.3   PROOF OF IDENTIFIABILITY THEORY

### A.3.1   PRELIMINARIES

We begin by introducing some related concepts and lemma essential for our proof.

**Definition A.1** (First-order Markov Property (Sutton & Barto, 2018)). A stochastic process $\{X_t : t \in \mathcal{N}\}$ has the first-order Markov property if, for each set of times $t, t-1, \ldots, 0$ and corresponding state $x_t, x_{t-1}, \ldots, x_0$ in the state space, the following conditional independence property holds:

$$\mathbb{P}(X_t = x_t | X_{t-1} = x_{t-1}, X_{t-2} = x_{t-2}, \ldots, X_0 = x_0) = \mathbb{P}(X_t = x_t | X_{t-1} = x_{t-1}) \quad (6)$$

The first-order Markov property implies that the transition probability to the next state depends only on the current state, uninfluenced by the sequence of previous states. In the context of state transition process, it possess the first-order Markov property. Mathematically, it can be represented as:

$$\mathbb{P}(\vec{s}_t | \vec{s}_{t-1}, \vec{a}_{t-1}, \vec{s}_{t-2}, \vec{a}_{t-2}, \ldots, \vec{s}_0, \vec{a}_0) = \mathbb{P}(\vec{s}_t | \vec{s}_{t-1}, \vec{a}_{t-1}), \quad (7)$$

where $\mathbb{P}(\vec{s}_t | \vec{s}_{t-1}, \vec{a}_{t-1})$ is the transition probability from $(\vec{s}_{t-1}, \vec{a}_{t-1})$ to the state $\vec{s}_t$.

**Definition A.2** (Finite Mixture Models (McLachlan et al., 2019)). A finite mixture model is a statistical model that assumes the presence of unobserved groups within a total population. In this model, the total population consists of a finite number of unobserved groups, each characterized by its own probability distribution.

In the finite mixture model framework, the model of the total population is constructed as a weighted sum of these individual distributions, providing a flexible framework for analyzing and interpreting the inherent heterogeneity within the population.

Following prior work, we define identifiability in representation function space.

**Definition A.3** (Identifiability). Let $\vec{s}_t$ be a sequence of observed variables generated by the true individualized transition processes specified by $(f_i, l, p_{\epsilon_i})$ given in the preliminary. A learned generative model $(\hat{f}_i, \hat{l}, \hat{p}_{\epsilon_i})$ is observationally equivalent to $(f_i, l, p_{\epsilon_i})$ if the joint distribution $p_{\hat{f}, \hat{l}, \hat{p}_\epsilon}(\vec{s}_t)$

matches $p_{f,l,p_\epsilon}(\vec{s}_t)$ everywhere. We say the latent individualized-specific factors are identifiable if observational equivalence can always lead to identifiability of the latent variables:

$$p_{\hat{f},\hat{l},\hat{p}_\epsilon}(\vec{s}_t) = p_{f,l,p_\epsilon}(\vec{s}_t) \Rightarrow \hat{l} = l. \tag{8}$$

### A.3.2 PROOF OF THEOREM 1

Theorem 1 shows the identifiability of the individual-specific factor $L$ in the individualized transition model. Suppose the sampled individuals are from different unobserved groups, and the transition dynamics are different across groups but identical within each group.

**Theorem 2.** *Assume the individualized transition processes in Eq. 2, where the nonlinear state transition functions $f_i$ are stationary within each individual but exhibit variability across different individuals. The individual-specific factor $L$ is a discrete variable, and the different values of $L = l_k$ delineate different transition processes determined by $f_i^k$ for each individual. Here we assume:*

- *(Group Determinacy): The individual-specific factor $L = l_k$ delineates distinct groups within the finite mixture model, each of which defines and dictates the individualized transition dynamics $f_i^k$.*

- *(Sample Sufficiency): The length of the sequence for each individual $l_s$ is greater than $2q - 1$, where $q$ is the number of groups.*

- *(Sequential Dependency): At any time $t$, $\vec{s}_t$ is determined exclusively by the given conditions $\subseteq (\vec{s}_{t-1}, \vec{a}_{t-1}, L)$ within each group, and there are no instantaneous relations between $\vec{s}_t$.*

*Then the identifiability property of the individual-specific factor $L$ is ensured.*

**Proof** We first show that the individualized transition processes can be viewed as a finite mixture model with grouped samples, then based on Lemma 1, the identifibility of latent individual-specific factor can be derived if the observed samples are sufficient.

*Finite Mixture Model Representation* The individualized transition process in Eq. 2 consists of a finite number of different processes $s_{it}$ (or mixture groups), each of which represents the behavior of a specific group within the population. Thus the individualized transitions $s_{it}^k$ are generated from different submodels $f_i^k$. According to the assumption of Group Determinacy, each $f_i^k$ is corresponding to a different component in the mixture model, characterized by different parameters or dynamics, and conditioned on the specific value of the latent variable $L = l_k$. According to the assumption of Sequential Dependency, the mixture component can be integrated and represented by:

$$\mathbb{P}(s_{it}|s_{it-1}, a_{it-1}) = \sum_{k=1}^{q} \mathbb{P}(s_{it}|s_{it-1}, a_{it-1}, L = l_k)\mathbb{P}(L = l_k) \tag{9}$$

$$\implies \quad \mathbb{P}(\vec{s}_t|\vec{s}_{t-1}, \vec{a}_{t-1}) = \sum_{k=1}^{q} \mathbb{P}(\vec{s}_t|\vec{s}_{t-1}, \vec{a}_{t-1}, L = l_k)\mathbb{P}(L = l_k) \tag{10}$$

where $\mathbb{P}(L = l_k)$ is the mixing coefficient of each group $k$, indicating the probability that a sample belongs to a particular group, and $l_k$ is considered as the individual-specific factors, indicating to which group $k$ a sample belongs and the strength of the individualized influence.

This likelihoood expression already has the form of a finite mixture model, where each term in the summation corresponds to a group of the mixture. Thus each observation represents a mixture component, which can represent different dynamic regimes of the transition process, and observations in the same group are known to be drawn from the same component. This aligns with the lemma that observations in the same group are known to be drawn from the same component.

*Lemma* Lemma 1 (Vandermeulen & Scott, 2015) addresses the identifiability of mixture models from grouped samples. It states that in the context of a mixture model, where the observations are segmented into $q$ distinct groups, and each group is guaranteed to come from the same component. If there are at least $2q - 1$ observations for each group, then each mixture of the $q$ probability measures incorporated in the model can be uniquely identified. This allows for a rigorous distinction of the individual components within the mixture model.

**Lemma 1.** *Suppose we have observations from a mixture model and that they are grouped, such that observations in the same group are known to be drawn from the same component. Denote by $q$ the number of groups. If there are at least $2q - 1$ observations per group, any mixture of $q$ probability measures can be uniquely identified.*

*Identifiability* Given the lemma, for the latent variable $L$ to be identifiable, there should be at least $2q - 1$ observations per group, where $q$ is the number of groups. Therefore, accroding to the Sample Sufficiency assumption, each individual in the transition process has at least $2q - 1$ transition tuples, then the latent variable $L$ would be identifiable according to the given lemma.

**Remark 1.** *In Huang et al. (2019), a Gaussian mixture model was used as a prior on the coefficients, while the latent confounder variable, denoted as $Z$, was constrained to a binary state, thereby indicating group membership for a given individual. In our work, we extend this foundation by generalizing the latent confounder to a broader set of discrete values. This extension to discrete categorization offers several advantages. First, it allows for a more computationally tractable methodology, making it easier to process and analyze the data. Second, it provides a versatile framework for capturing different types of heterogeneity that may exist within the sample population. In particular, the discretized nature of the latent variable $L$ enhances the interpretability of the model. It allows the understanding of the underlying patterns and effects manifested by the latent confounder in a more nuanced way.*

## B    EXTENDED BACKGROUND

**Reinforcement Learning**    In RL, an agent learns to make decisions by interacting with the environment. The agent receives rewards for taking actions in the environment and uses this feedback to learn optimal behavior. It is often modeled as a Markov Decision Process (MDP) represented by a tuple $\langle \mathcal{S}, \mathcal{A}, \mathbb{P}, R, \gamma \rangle$, where $\mathcal{S}$ denotes a finite set of states representing different situations an agent might encounter, $\mathcal{A}$ a finite set of actions representing different decisions an agent can make, $\mathbb{P}$ a state transition function defining the probability of transitioning to a new state $s'$ given a current state $s$ and action $a$, denoted as $\mathbb{P}(s'|s,a)$, $R$ a reward function assigning a scalar value to each state-action pair $(s, a)$, representing the immediate reward received after performing action $a$ in state $s$. $\gamma \in [0, 1]$ is the discount factor, representing the agent's consideration for future rewards. The agent's goal is to learn an optimal policy $\pi^*$, which defines the optimal set of actions in different states to maximize the expected cumulative discounted reward over the long run. Developing this optimal policy involves estimating value functions such as the action-value function, defined as $Q^\pi(s, a) = \mathbb{E}_\pi \left[ \sum_{t=0}^\infty \gamma^t R_t | S_0 = s, A_0 = a \right]$, which represents the expected reward of taking action $a$ in state $s$ following policy $\pi$. The pursuit of optimal policy $\pi^*$ involves maximizing the value functions over all possible state-action pairs: $\pi^* = \arg\max_\pi Q^\pi(s, a)$.

**Model-based Reinforcement Learning**    Model-based Reinforcement Learning (MBRL) is a branch of RL where an explicit model of the environment's dynamics is either known prior or learned through interaction with the environment. This model is then used for planning and decision-making to optimize the agent's policy. In the MBRL framework, the learning process is bifurcated into two components: learning a model of the environment's dynamics and using this model to make decisions. In the first component, the agent interacts with the environment and collects data in trajectories. A trajectory refers to a sequence of state-action-reward tuples an agent experiences while interacting with the environment. A single trajectory with reward is often represented as $\tau = \{(\vec{s}_0, \vec{a}_0, r_0), (\vec{s}_1, \vec{a}_1, r_1), \ldots, (\vec{s}_T, \vec{a}_T, r_T)\}$, where $\vec{s}_t$ and $\vec{a}_t$ represents the vector of states and actions at time $t$ respectively, and $r_t$ denotes the immediate reward post-action. $T$ is the final time step of the trajectory, which can be the terminal state of the episode or the set horizon. After that, the collected trajectories are used to learn or refine the model of the environment by approximating the transition dynamics $\mathbb{P}(s'|s,a)$. This model is essential in planning, simulating possible trajectories, and evaluating action sequences to identify optimal actions based on cumulative rewards. Once the model is learned, it is used for planning by exploring hypothetical trajectories and optimizing action selection in unexplored states and situations, enhancing overall task performance. This method facilitates efficient exploration of the state-action space, minimizing necessary interactions with the environment, a vital feature when such interactions are costly or risky. The accuracy of the model is crucial, as it directly influences the agent's ability to make informed decisions and adapt to new scenarios, ultimately improving the efficiency and efficacy of the learning process.

**Variational Autoencoder** Variational Autoencoders (VAEs) are a class of generative models in deep learning, adept at unsupervised learning of complex data distributions. Rooted in the framework of Bayesian inference, VAEs are designed to approximate probability density functions of input data. The architecture of a VAE consists of two primary components: an encoder $q_\phi(z|x)$ and a decoder $p_\theta(x|z)$. The encoder maps input data $x$ to a latent space, represented by a probability distribution, typically Gaussian, with parameters $\mu$ and $\sigma$ signifying the mean and standard deviation, respectively. The decoder reconstructs the input data from a sampled latent representation $z$.

The distinct feature of VAEs lies in their probabilistic approach. The encoder outputs parameters of a latent distribution, from which a sample $z$ is drawn:

$$z \sim q_\phi(z|x) = \mathcal{N}(z; \mu, \sigma^2 I) \tag{11}$$

The decoder then attempts to reconstruct the input from this latent sample. VAEs optimize the Evidence Lower Bound (ELBO) objective, which balances two aspects: the reconstruction quality and the regularization of the latent space. The ELBO is given by:

$$\text{ELBO} = \mathbb{E}_{q_\phi(z|x)}[\log p_\theta(x|z)] - \text{KL}[q_\phi(z|x)||p(z)] \tag{12}$$

Here, the first term measures the reconstruction fidelity, while the second term, the Kullback-Leibler (KL) divergence, imposes a regularization by encouraging the latent distribution $q_\phi(z|x)$ to be close to a prior $p(z)$, typically assumed to be a standard normal distribution $\mathcal{N}(0, I)$. VAEs, through this optimization, are capable of generating new data points that are similar to the input data, making them highly valuable in applications like image generation, denoising, and anomaly detection within the domain of unsupervised learning.

## C    DETAILED RELATED WORK

**Individualized Machine-Learning Applications** In the modern era, the power of machine learning has been harnessed to create highly individualized solutions across a myriad of domains. In the realm of health and wellness, machine learning aids in tailoring interventions for increasing physical activity (Yom-Tov et al., 2017; Liao et al., 2020), promoting weight loss (Forman et al., 2019; 2023), improving adherence for diabetes (Yom-Tov et al., 2017). For the elderly, personalized algorithms assist in both technology adaptation and specialized care for conditions (Hoey et al., 2014). The financial sector benefits from machine learning's prowess in optimizing technical indicators, making stock market predictions more precise and individualized (Li et al., 2019). In the educational landscape, Information and Communication Technology (ICT) leverages machine learning to offer personalized education systems such as adaptive e-learning interfaces (Fok et al., 2005) and individualized tutorial planning (Ji et al., 2017). Furthermore, the transportation sector sees advancements with car-following control strategies tailored for individual drivers (Song et al., 2023). Multimedia platforms, such as YouTube and TikTok, are enhancing user experiences by offering video content recommendations fine-tuned to individual preferences using reinforcement learning (Cai et al., 2022; Hoiles et al., 2020). These examples merely scratch the surface, emphasizing the vast and diverse applications of individualized machine learning in today's world.

**Reinforcement Learning for Latent State-Transition Processes** RL has witnessed significant advancements in recent years, particularly with the integration of latent variable models to capture the underlying dynamics of environments. A primary focus in this domain is learning low-dimensional, latent Markovian representations from observed data (Lesort et al., 2018; Krishnan et al., 2015; Karl et al., 2016; Ha & Schmidhuber, 2018; Watter et al., 2015; Zhang et al., 2018; Kulkarni et al., 2016; Mahadevan & Maggioni, 2007; Gelada et al., 2019; Gregor et al., 2018; Ghosh et al., 2019; Zhang et al., 2021). Common strategies for state representation learning include reconstructing the observation, learning a forward model, or learning an inverse model. Additionally, prior knowledge, such as temporal continuity (Wiskott & Sejnowski, 2002), can be leveraged to constrain the state space. Numerous studies have proposed methods to estimate the underlying state-transition process from high-dimensional input sequences (Watter et al., 2015; Ebert et al., 2017; Ha & Schmidhuber, 2018; Hafner et al., 2018; Zhang et al., 2019; Gelada et al., 2019; Kaiser et al., 2019; Hafner et al., 2020). Using the learned world model, agents can engage in model-based RL or planning. Furthermore, these methods encode structural constraints, ensuring the sufficiency and minimality of the estimated state representations from both generative and selection processes. Recently, several studies (Lu et al., 2018; Li et al., 2020; Vo et al., 2021; Wang et al., 2021; Bennett et al., 2021; Pace et al.,

2023) have aimed to estimate the state-transition process in the presence of latent confounders. A handful of work (Lu et al., 2018; Pace et al., 2023) can be viewed as addressing similar settings involving individual-specific factors. However, to the best of our knowledge, we have yet to identify a systemic approach that offers a clear identifiability result for the state-transition process when individual-specific factors are present.

## D  ALGORITHM

The pseudocode for the proposed algorithm is presented in Algorithm 1 and Algorithm 2.

## E  EXTENDED EXPERIMENT

### E.1  EVALUATION METRICS

**Identifibility Metric**    The Pearson Correlation Coefficient (PCC) Cohen et al. (2009) is a statistical measure that quantifies the degree of linear relationship between two variables. It provides a value between -1 and 1, where 1 implies a perfect positive linear relationship, -1 implies a perfect negative linear relationship, and 0 implies no linear relationship between the variables. The equation for calculating the Pearson Correlation Coefficient $r$ between two variables $X$ and $Y$ is as follows:

$$r = \frac{n(\sum xy) - (\sum x)(\sum y)}{\sqrt{[n \sum x^2 - (\sum x)^2][n \sum y^2 - (\sum y)^2]}} \tag{13}$$

where $n$ is the number of paired samples, $\sum xy$ is the sum of the product of paired scores, $\sum x$ and $\sum y$ are the sums of the $x$ scores and $y$ scores respectively, $\sum x^2$ and $\sum y^2$ are the sums of the squared $x$ scores and $y$ scores respectively.

Canonical Correlation Analysis (CCA) (Hotelling, 1992) is designed to identify bases for two sets of variables in order to maximize the mutual correlations between the projections onto these bases. In our work, CCA is used as an evaluation metric to validate that the recovered latent variable is meaningfully related to the ground truth latent variable, thus proving the relevance of the estimated representations. Let $X$ and $Y$ be the two sets of observed variables. This algorithm starts by centering the columns of $X$ and $Y$ so that they have zero mean. Then the covariance matrices $C_{XX} = X\top X, C_{YY} = Y\top Y$, and $C_{XY} = X\top Y$ are calculated. After that, the canonical correlations are obtained by solving the following generalized eigenvalue problem: $C_{XX}^{-1} C_{XY} C_{YY}^{-1} C_{YX} \nu = \lambda \nu$. The square roots of the eigenvalues $\lambda$ indicate the canonical correlations between the linear combinations of $X$ and $Y$. The corresponding eigenvectors $\nu$ and $u = C_{XY}\nu$ are the canonical weights used to construct the canonical variables. Finally, the canonical variables of $X$ and $Y$ are $U = X\nu$ and $V = Yu$, respectively, representing the linear combinations of the original variables that are maximally correlated. The correlation of the primary pair of canonical variables is the highest, followed by the secondary pair, and so on. When employing CCA as an evaluation metric, a higher canonical correlation indicates a stronger and more relevant relationship between the recovered latent variable and the ground truth latent variable.

### E.2  DATASET DESCRIPTIONS

**Synthetic Data Generation Processes**    In this paper, we created three synthetic datasets: One with a single discrete latent variable that satisfies our assumptions and two others that violate our assumptions, allowing for multiple discrete and continuous latent variables. The dimensions of states and actions are set to 3 and 2, respectively. The initial state and the actions taken are generated randomly, following a uniform distribution $\text{Uniform}(0, 1)$. The noise term is modeled to follow a Gaussian distribution with zero mean and latent-dependent variance. The mixing function $f$ conforms to the post-nonlinear model (Zhang & Hyvarinen, 2012), where $f_1$ represents the nonlinear effect, and $f_2$ denotes the invertible post-nonlinear distortion on $\vec{s}_t$, embodied by a randomly initialized three-layer MLP with tanh units. The data generation processes are modeled as follows:

$$\vec{s}_t = f_2(f_1(\vec{s}_{t-1}, \vec{a}_{t-1}, \vec{L}), \epsilon_t)). \tag{14}$$

- *Single Discrete Latent Dataset*. We generate datasets consisting of 100 individuals, typically divided into four distinct groups, with each individual sequence having a total length of 300. The

---

**Algorithm 1** Algorithm of Individualized Policy Learning through Latent Factor Estimation

---

**Require:** $\{f_{\text{Env}}^k\}_{k=1}^K$: individualized environments; En: encoder; Quantize: embedding dictionary; Noise: noise estimator; De: decoder; $\pi$: policy network

**Ensure:** $\hat{l}_k$: estimated individual-specific factor; $\{\pi_k^*\}_{k=1}^K$: individualized policy

1: **procedure** MAIN($f_{\text{Env}}$, En, Embed, Noise, De, $\pi$)                                       ▷ main loop for IPLF
2:     En, Embed, Noise, De, $\pi \sim \text{N}(0, \text{I})$                                       ▷ randomly initialize the network
3:     $\mathcal{H} \leftarrow \{\tau_k\}_{k=1}^K$          ▷ collect individual trajectory samples by interaction with $\{f_{\text{Env}}^k\}_{k=1}^K$
4:     **for** each individual trajectory $k$ **do**
5:         $z^k = \text{Encoder}(\vec{s}_{0:T}^{\,k})$                                       ▷ capture the unique individual representations
6:         $\hat{l}_k = \text{Quantize}(z^k)$                                       ▷ vector quantization
7:         $\hat{\nu}_k = \text{Noise}(\hat{l}_k)$                                       ▷ estimate noise with MLP layer
8:         **for** each next state $\vec{s}_t^{\,k}$ in the trajectory **do**
9:             $\hat{\vec{s}}_t^{\,k} = \text{Decoder}(\vec{s}_{t-1}^{\,k}, \vec{a}_{t-1}^k, \hat{l}_k, \hat{\nu}_k)$                                       ▷ reconstruct the next state
10:         **end for**
11:     **end for**
12:     **return** $\{\pi_k^*\}_{k=1}^K = \text{PolicyLearning}(\mathcal{H}, \{\hat{l}_k\}_{k=1}^K)$          ▷ optimize the individualized policies
13: **end procedure**

14: **function** ENCODER($\vec{s}_{0:T}^{\,k}$)
15:     **if** dataset is synthetic **then**
16:         **for** each $t$ in $\vec{s}_{0:T}^{\,k}$ **do** $o_t^k \leftarrow \text{Conv1D}(\vec{s}_{t:t+H}^{\,k})$
17:         **end for**
18:     **else if** dataset is corpus **then**
19:         Initialize $h_0^k, c_0^k$
20:         **for** each $t$ in $\vec{s}_{0:T}^{\,k}$ **do** $h_t^k, c_t^k \leftarrow \text{LSTM}(h_{t-1}^k, c_{t-1}^k, \vec{s}_t^{\,k}; \theta)$
21:         **end for**
22:     **end if**
23:     **return** $z^k \leftarrow$ Final output of Conv1D or final hidden state of LSTM
24: **end function**

25: **function** QUANTIZE($z^k$)
26:     Initialize $E = \{e_1, e_2, \ldots, e_q\}, d_{\min} = \infty$
27:     **for** each $e_i$ in $E$ **do**
28:         **if** $\|z^k - e_i\|_2 < d_{\min}$ **then**
29:             Update $d_{\min}$ and $\hat{l}_k \leftarrow e_i$
30:         **end if**
31:     **end for**
32:     **return** $\hat{l}_k$
33: **end function**

34: **function** DECODER($\vec{s}_{t-1}^{\,k}, \vec{a}_{t-1}^k, \hat{l}_k, \hat{\nu}_k$)
35:     Combine inputs to reconstruct $\hat{\vec{s}}_t^{\,k} \leftarrow \text{Decoder}(\vec{s}_{t-1}^{\,k}, \vec{a}_{t-1}^k, \hat{l}_k, \hat{\nu}_k)$
36:     **return** Reconstructed state $\hat{\vec{s}}_t^{\,k}$
37: **end function**

38: **function** POLICYLEARNING($\mathcal{H}, \{\hat{l}_k\}_{k=1}^K$)
39:     **for** each individual $k$ **do**
40:         Update policy input to $\mu_\pi(\vec{s}_t; \theta^\mu) \rightarrow \mu_\pi^K(\vec{s}_t^{\,k}, \hat{l}^k; \theta^\mu)$
41:         Update training objective:
42:             $J(\theta^\mu) = \mathbb{E}\left[\sum_{t=0}^\infty \gamma^t Q\left(\vec{s}_t, \mu_\pi^K(\vec{s}_t^{\,k}, \hat{l}_k; \theta^\mu); \theta^Q\right)\right]$
43:         Optimize $\mu_\pi^K$ for individual $k$
44:     **end for**
45:     **return** Updated policy $\mu_\pi^K$
46: **end function**

---

**Algorithm 2** Training Process with Extended ELBO Objective

---

1: Initialize parameters of the encoder Encoder and decoder Decoder
2: Initialize weights $\alpha$ and $\beta$
3: **repeat**
4:     **for** each individual $k$ **do**
5:         Compute encoded representation: $z^k \leftarrow \text{Encoder}(\vec{s}^k_{0:T})$
6:         Estimate individual-specific factor: $\hat{l}_k \leftarrow \text{Quantize}(z^k)$
7:         Compute reconstructed state: $\hat{\vec{s}}^k_t \leftarrow \text{Decoder}(\vec{s}^k_{t-1}, \vec{a}^k_{t-1}, \hat{l}_k, \hat{\nu}_k)$
8:         Calculate $\mathcal{L}_{\text{Recon}} = \sum_t \|\vec{s}^k_t - \hat{\vec{s}}^k_t\|^2$
9:         Calculate $\mathcal{L}_{\text{Quant}} = \sum_i \|\text{sg}[z^k_i] - e^k_i\|^2$, $\mathcal{L}_{\text{Commit}} = \sum_i \|e^k_i - \text{sg}[z^k_i]\|^2$
10:         Compute extended ELBO objective: $\mathcal{L}_{\text{ELBO}} = \mathcal{L}_{\text{Recon}} + \beta\mathcal{L}_{\text{Quant}} + \alpha\mathcal{L}_{\text{Commit}}$
11:         Update parameters to minimize $\mathcal{L}_{\text{ELBO}}$
12:     **end for**
13: **until** convergence

---

single discrete latent follows the categorical distribution $\text{Cat}(0.1, 0.2, 0.3, 0.4)$ with cardinality 4. The noise $\epsilon^k_{it}$ is sampled from an i.i.d. Gaussian distribution with its variance modulated by four different latent values. A 2-layer MLP with ReLU units is used to generate the latent-dependent variance of the noise.

- *(Violation) Multiple Discrete Latents Dataset.* We generate datasets consisting of 100 individuals, typically divided into twenty-four distinct groups, with each individual sequence having a total length of 300. The multiple independent discrete latents have a dimensionality of three, which follow the categorical distributions $\text{Cat}(0.1, 0.9)$, $\text{Cat}(0.2, 0.3, 0.5)$ and $\text{Cat}(0.1, 0.2, 0.3, 0.4)$ with cardinality 2, 3 and 4, respectively. The noise $\epsilon^k_{it}$ is sampled from an i.i.d. Gaussian distribution with the variance being modulated by the latent vectors. A 2-layer MLP with ReLU units is used to generate the latent-dependent variance in the noise.

- *(Violation) Multiple Continous Latents Dataset.* We generate datasets consisting of 100 individuals with each individual sequence having a total length of 300. The multiple independent discrete latents have a dimensionality of three, which follow the Gaussian distribution $\mathcal{N}(0, 1)$, uniform distribution $\text{Uniform}(0, 1)$, and exponential distribution $\text{Exp}(1)$. The noise $\epsilon^k_{it}$ is sampled from an i.i.d. Gaussian distribution with the variance being modulated by the latent vectors. A 2-layer MLP with ReLU units is used to generate the latent-dependent variance in the noise.

**Persuasion For Good**   The Persuasion For Good Corpus presents a valuable collection of online dialogues carefully curated by participants from Amazon Mechanical Turk. In each interaction, one participant, referred to as the *persuader*, attempts to persuade the other participant, referred to as the *persuadee*, to contribute to a charitable cause. This comprehensive dataset includes 1,017 such conversations, all accompanied by additional demographic data and psychological survey responses from the participants involved. For each utterance in the dataset, corresponding to a turn in a dialogue, the following attributes are provided in Table 2. Fortunately, all the participants underwent personality assessments, allowing us to use the labeled 35-dimensional individualized personalities of each persuadee as ground-truth latents in our experiments.

Table 3 displays excerpts from the Persuasion for Good Corpus Dataset. In our experiment, we use a pre-trained BERT model to convert each dialogue into vectors. The process starts with tokenization, breaking words into smaller units. These tokens are then input into BERT, which provides contextual embeddings for each token, considering the entire dialogue context.

**Pendulum Control**   The pendulum environment, provided by OpenAI Gym, is a classic control task used for the evaluation and development of RL models. This environment presents a continuous control task where the agent must learn to control a frictionless pendulum with the goal of swinging it to the highest point and keeping it in the inverted position. The pendulum starts at a random position, and the goal is to bring it to a standstill at the inverted position with the least amount of effort. The system is characterized by a continuous action space, representing the torque applied to the pendulum's fulcrum. For a pendulum of length $l$ and mass $m$, subject to gravity $g$ and a control input $u$, the equations of motion can be described by the following second-order nonlinear ordinary

Table 2: Attribute descriptions in persuasion for good corpus.

| Attribute | Explanation |
| --- | --- |
| id | index of the utterance |
| speaker | author of the utterance |
| conversation_id | id of the first utterance in the dialogue in which the current utterance belongs to |
| reply_to | id of the prior utterance, or None if the current utterance starts the conversation |
| text | content of the utterance |
| role | whether the author of the utterance is the persuader (0) or persuadee (1) |
| user_turn_id | turn index of the user. i represents the user's ith turn in the conversation |
| label_1 | a list of dialogue acts in each sentence of the utterance |
| label_2 | the second dialogue act in each sentence, available for a limited number of utterances |
| sentiment | sentiment scores for each sentence, categorized as positive, negative, or neutral |
| n_sents | the total number of sentences in the utterance |
| text_by_sent | a string containing the utterance's text, where the ¡s¿ denotes sentence breaks |

Table 3: Excerpts from the Persuasion for Good Corpus Dataset.

| Speaker | Utterance |
| --- | --- |
| persuader | Hi, how are you today? |
| persuadee | I am fine. And you? |
| persuader | Not too bad. Have you heard of Save The Children? |
| persuadee | I have, actually. |
| persuader | They do great work at least I think what about you? |
| persuadee | I'm often skeptical of big charities like that. They sometimes don't put the money that is donated into the right projects. |

differential equations:

$$\dot{\theta} = \omega,$$
$$\dot{\omega} = -\frac{g}{l}\sin(\theta) + \frac{u}{ml^2},$$

where $\theta$ is the angle of the pendulum from the vertical upright position, and $\omega$ is the angular velocity of the pendulum. The state of the pendulum at any time $t$ can be represented as $\vec{s}_t = [\cos\theta_t, \sin\theta_t, \omega_t]$, action represents the torque applied to the free end of the pendulum in the range $a_t \in [-2, 2]$, and the reward function is defined as: $r_t = -(\theta_t^2 + 0.1 * \omega_t^2 + 0.001 * a_t^2)$. The goal of RL algorithms in this environment is to determine an optimal control policy $\pi^*$ that minimizes the effort to swing and balance the pendulum upright, typically by minimizing a cost function defined over states and actions. Each episode provides a continuous stream of observations, actions, and rewards, allowing the development and evaluation of algorithms capable of learning effective control policies in continuous action spaces. In academic studies, the Pendulum environment serves as a benchmark to investigate the effectiveness of RL algorithms in handling continuous control tasks.

## E.3 ADDITIONAL EXPERIMENT RESULTS

In the main paper, we set the number of individuals as 100 to demonstrate the effectiveness of our proposed method. Here, we will vary the number of samples to further verify this effectiveness. The data generation process is consistent with Case 1, except that we change the number of samples to $\{100, 150, 200, 300, 500, 800, 1000\}$. The comparison results shown in Figure 5 indicate that our method can achieve consistently good recovery performance under different numbers of individuals. This further confirms that the identifibility of our framework is guaranteed by the mathematical relationship between the trajectory length and the number of groups, which is constrained by the sample sufficiency assumption; the number of individuals is of lesser importance.

## F ESTIMATION FRAMEWORK

The proposed framework is meticulously customized based on the requirements of the identifiability theorem given in Section 4.3. Given the offline trajectory of the population, the individual-specific

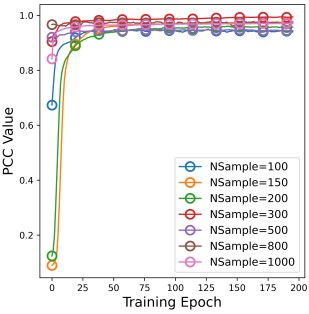

Figure 5: PCC trajectory comparisons under different number of individuals.

factor estimation framework is developed to recover the latent confounder. To satisfy the group determinacy assumption, we first feed the entire sequence $\bar{s}_{0:T}^k$ of each individual $k$ into a quantized encoder to estimate the latent confounder $\hat{l}_k$. Then to compensate for the determinacy of the quantization operation, we introduce the noise estimator to increase the variability and robustness of the generated samples. To reconstruct individualized transition processes, we design a conditional decoder that takes the conditions $(\bar{s}_{t-1}^k, \vec{a}_{t-1}^k)$, the estimated confounder $\hat{l}_k$, and the estimated noise $\hat{\nu}_k$ as inputs, with $\hat{\bar{s}}_t^k$ as the output. The optimized quantized encoder is used to augment the original buffer and facilitate the downstream individualized policy learning.

The traditional VAE (Kingma & Welling, 2013) encodes input data into a continuous latent space using probabilistic encoders, and then reconstructs the input from this space using decoders. While our proposed framework differs from the general continuous VAE and model-based RL in four main aspects: 1) Our framework uses a quantization layer to discretize the continuous latent representations. This mapping of continuous latent representations to an embedding dictionary is well suited to the group determinacy requirement. 2) We estimate noise as an input to the decoder to introduce variability that satisfies our individualized transition processes and allows the model to better vary the data. 3) Our decoder reconstructs individualized transition processes to simulate the data generation process, incorporating additional conditions, estimated latent individual-specific factor, and noise. 4) In addition to the original population data, we further extract latent factors for each individual as additional information to facilitate the individual policy learning.

**Encoder** As for the Conv1D, let the Conv1D layer transform an input sequence $\bar{s}_t^k$, using learned kernel filters. These filters slide over the sequence to produce a feature map, denoting the response of the filter at each position. Mathematically, the transformation by a single filter in the Conv1D layer at time $t$ is described as:

$$\vec{o}_t = \sigma \left( W * \bar{s}_{t:H+t}^k + b \right),$$

where $\vec{o}_t$ is the feature map at time $t$, $W$ the kernel to be learned during training, $*$ the convolution operation, $\bar{s}_{t:H+t}^k$ the input sub-sequence from time $t$ to $t + H$, where $H$ is the size of the kernel. $\sigma$ is the activation function, and $b$ is the bias term to be learned during training. The layer may contain multiple such filters, each learning different features of the input sequence. The resulting feature maps serve as a transformed representation $z^k$, which embeds the information about the latent individual-specific factor $L$.

As for the LSTM, let the hidden states and cell states of the LSTM at time $t$ denote as $h_t$ and $c_t$, respectively. Then, the LSTM updates are given by:

$$\begin{aligned}
f_t &= \sigma(W_f \cdot [\bar{s}_t^k, h_{t-1}] + b_f), \\
i_t &= \sigma(W_i \cdot [\bar{s}_t^k, h_{t-1}] + b_i), \\
\tilde{c}_t &= \tanh(W_c \cdot [\bar{s}_t^k, h_{t-1}] + b_c), \\
c_t &= f_t \odot c_{t-1} + i_t \odot \tilde{c}_t, \\
o_t &= \sigma(W_o \cdot [\bar{s}_t^k, h_{t-1}] + b_o), \\
h_t &= o_t \odot \tanh(c_t),
\end{aligned}$$

where $\sigma$ represents the sigmoid activation function, $\odot$ element-wise multiplication. $W_f, W_i, W_c, W_o$ and $b_f, b_i, b_c, b_o$ are the weight matrices and bias terms to be learned during training. $f_t, i_t, \tilde{c}_t, c_t, o_t$ and $h_t$ are the forget gate, input gate, candidate cell state, cell state, output gate, and hidden state at time $t$ respectively. The final hidden state of the LSTM $h_T$, after the sequential processing of the entire trajectory, serves as the representative $z^k$ that embeds the information about the latent individual-specific factor $L$.

**Quantization Layer**  Let the output of the encoder be a continuous latent representation denoted as $z^k \in \mathbb{R}$, and define an embedding dictionary $E$ consisting of $q$ vectors, where each vector represents a unique discrete category: $E = \{e_1, e_2, \ldots, e_q\}$, where $e_i \in \mathbb{R}$. The quantized vector $\hat{l}_k$ is obtained by mapping $z^k$ to the nearest dictionary vector. The mapping can be expressed mathematically as: $\hat{l}_k = \arg\min_{e_i \in E} \|z^k - e_i\|_2$. Subsequently, the quantized output is the vector from the dictionary that is closest to the encoder output. Thus, the continuous representation $z^k$ is effectively mapped to a discrete $\hat{l}_k$ by finding the nearest neighbor in the dictionary, aligning the representation learning with the discrete nature of the latent variable.

**Decoder**  Define $\vec{s}_{t-1}^k$ and $\vec{a}_{t-1}^k$ as the true previous state and action, respectively, and let $\hat{\nu}_k$ be the estimated noise term. Additionally, let $\hat{l}_k$ be the approximated latent individual-specific factor for the $k$-th individual. The inputs to the conditional decoder at time $t$ are a combination of the aforementioned variables: $\text{Input}_t = (\vec{s}_{t-1}, \vec{a}_{t-1}, \hat{\nu}_k, \hat{l}_k)$. The output of the decoder is the reconstructed next state, $\hat{\vec{s}}_t$, which is a function of the decoder input: $\hat{\vec{s}}_t = \text{De}(\text{Input}_t)$. The reconstruction likelihood measures how closely the reconstructed state matches the true subsequent state, which is defined as $\mathcal{L}_{\text{Recon}} = p_{\text{Recon}}(\vec{s}_t^k | \vec{s}_{t-1}^k, \vec{a}_{t-1}^k, \hat{\nu}_k, \hat{l}_k)$. The objective in this process is to optimize the decoder parameters to maximize the reconstruction likelihood $\max \mathcal{L}_{\text{Recon}}$ so that the reconstructed state $\hat{\vec{s}}_t$ is as close as possible to the true next state $\vec{s}_t$.