# OpenReview forum: "Identifying Latent State Transition Processes for Individualized Reinforcement Learning"
_ICLR.cc/2024/Conference — Submitted to ICLR 2024_

### Official Review · Reviewer_Mxvj · 2023-10-27

**Soundness:** 3 good
**Presentation:** 2 fair
**Contribution:** 3 good
**Rating:** 6
**Confidence:** 3

**Summary:**

This paper mainly focuses on the problem of identifying the individual-specific state-transition processes. To handle this problem, this paper introduces a method that proficiently learns these processes from observed state-action trajectories and individualized policies. The results of synthetic and real-world datasets seem to demonstrate the effectiveness of the proposed methods.

**Strengths:**

1. This paper is well-written and easy to read.
2. The contribution of the paper is clear, and the proposed method is relatively novel.
3. The code has been provided, which increases the reproducibility.

**Weaknesses:**

The paper has a few explanation of why each part can work. It may be more helpful for readers to understand if the effect of each component and why it works can be more clearly displayed.

**Questions:**

Due to the discussion of Individualized Markov Decision in the paper, Processes does not add a reference, which is a custom task for the paper. So, is there any significant difference between L (individualized latent space) and state in IMDP? Are there any significant differences between the two? Can L be regarded as an unobserved state?

---

> ### Author Response · Authors · 2023-11-19
> **Rebuttal by Authors [Part 1]**
>
> **Q1: The paper has a few explanation of why each part can work. It may be more helpful for readers to understand if the effect of each component and why it works can be more clearly displayed.**
>
> **A1**: Thanks for your constructive comments. We have carefully revised the manuscript to clearly explain the function and importance of each component. Specifically, we have improved Section 4 by adding the following:
>
> 1) An overview paragraph that outlines the design of our methodology and provides a brief summary of the overall framework.
>
> > Our proposed framework is carefully customized to meet the requirements of the identifiability theorem. According to the identifiability definition A.3, latent confounders are identifiable if observational equivalence can consistently lead to the identifiability of the latent variables. This motivates our work to utilize a generative model and achieve confounder estimation by fitting the learned distribution to the true observed distribution. Traditional continuous VAE~\citep{kingma2013auto} encodes the input data into a continuous latent space using probabilistic encoders and then reconstructs the input from this space using decoders. However, this approach is not suitable for our problem setting. Specifically, given the offline trajectory of the population, we have customized the VAE framework and developed an individual-specific factor estimation framework to recover the latent factors.
>
> > First, to obtain latent representations in a reduced-dimensional space, we use an encoder with the entire sequence $\vec{s}^k\_{0:T}$ of each individual $k$ as input and the latent continuous representation $z^k$ as output. Considering the group determinacy assumption, the estimated latent individual-specific factor should be discrete. We then incorporate a quantization layer to discretize the continuous latent representation $z^k$ into the latent individual-specific factor $\hat{l}\_k$. Intuitively, the better $\hat{l}\_k$ is recovered, the more effectively the data generation process can be learned. To reconstruct individualized transition processes, we design a conditional decoder that takes the conditions $(\vec{s}^k\_{t-1},\vec{a}^k\_{t-1})$, the estimated latent individual-specific factor $\hat{l}\_k$, and the noise term $\hat{\nu}\_k$ as inputs, producing $\hat{\vec{s}}^k\_t$ as the output. The noise is estimated based on $\hat{l}\_k$ to account for the stochasticity of the state transition processes. Afterward, the optimized quantized encoder is employed to augment the original buffer and facilitate downstream individualized policy learning.
>
> 2) A straightforward explanation of the functions of each component, including its inputs/outputs and essential mathematical formulations. Specifically,
>
> -  In the quantized encoder module, we have added the motivation for introducing the quantization layer, along with corresponding update functions. The changes are as follows:
>
> > The group determinacy assumption implies the existence of the latent individual-specific factor $L$. Since $L$ denotes the time-invariant latent individual-specific factors that influence each state in the transition process, we initially use an encoder to capture the continuous representation $z^k$ from $\vec{s}^k\_{0:T}$ and then employ a quantization layer to define unique and distinct groups as $\hat{l}^k$.
>
> > The corresponding update functions are$o^k\_t=\text{Conv1D}(\vec{s}^k\_{t:t+H})$, $h^k\_t,c^k\_t=\text{LSTM}(h^k\_{t-1},c^k\_{t-1},\vec{s}^k\_t;\theta)$, where $o^k\_t$, $h^k\_t$ and $c^k\_t$ are the feature map, hidden state and cell state at time $t$, and $\theta$ denotes all the parameters of the LSTM.
>
> -  In the noise estimation module, we have provided the motivation for including this module and explained how it applies within our framework.
>
> > To account for the stochastic nature of the individualized transition processes, a noise estimator is introduced with $\hat{l}^k$ as input and $\hat{\nu}_k$ as output.
>
> > The motivations for introducing a noise layer are 1) to mimic the probabilistic property of the individualized state transition in Equation 1, and 2) to compensate for the loss of stochasticity due to the deterministic nature of the quantization operation.
>
> > Our posterior $ q(\hat{l}\_k=\kappa|\vec{s}^k_{0:T}) = \\{1, \text{for } \kappa=\arg\min\_j \lVert z^k-e\_j\lVert\_2; 0\text{ otherwise.}\\}$ is a categorical distribution and becomes deterministic after the quantization operation. By adding noise, we reintroduce variability that allows the model to better varying data, thus improving the model's ability to generalize and increase robustness in different scenarios.

---

> ### Author Response · Authors · 2023-11-19
> **Rebuttal by Authors [Part 2]**
>
> **Q2: Is there any significant difference between L and state in IMDP? Can L be regarded as an unobserved state.**
>
> **A2**: Mathematically, our $L$ is a latent confounder of all states. All existing work for MDP with latent variables have not considered this specfical case. Existing results cannot be directly applied. The details are as follows:
>
> - The state $ S_t $ usually refers to the specific situation of environment where the agent is present at timestep $ t $, and it only directly influences the subsequent state $ S_{t+1} $.
>
> - In contrast, $ L $  directly influences different states across all timesteps throughout the transition process.  $ L $ remains constant over time for each individual, while it may vary between individuals.  Mathematically, it is a latent confounder of all states.
>
> To the best our knowledge, we failed to find any works discuss such a latent confounder in the Markov decision process. Therefore existing work for MDP with latent variables cannot be directly applied here. These methods typically require that latent variables change over time [1-4]. The identifiability of having a latent confounder L of all states has not been explored in previous research.
>
>
>
> [1] Zhang, Xuezhou, et al. "Efficient reinforcement learning in block MDPs: A model-free representation learning approach." International Conference on Machine Learning. PMLR, 2022.
>
> [2] Guo, Zhaohan Daniel, et al. "Bootstrap latent-predictive representations for multitask reinforcement learning." International Conference on Machine Learning. PMLR, 2020.
>
> [3] Delgrange, Florent, Ann Nowe, and Guillermo A. Pérez. "Wasserstein Auto-encoded MDPs: Formal Verification of Efficiently Distilled RL Policies with Many-sided Guarantees." arXiv preprint arXiv:2303.12558 (2023).]
>
> [4] Zhang A, Lyle C, Sodhani S, et al. Invariant causal prediction for block mdps[C]//International Conference on Machine Learning. PMLR, 2020: 11214-11224.

---

> > ### Comment · Reviewer_Mxvj · 2023-11-21
> >
> > Thank you to the authors for their detailed response. I will keep my score.

---

### Official Review · Reviewer_y8Yq · 2023-11-01

**Soundness:** 2 fair
**Presentation:** 3 good
**Contribution:** 2 fair
**Rating:** 5
**Confidence:** 3

**Summary:**

The paper aims to develop individualized policy learning and adapt a policy to an upcoming individual, by considering individualized state-transition processes. The identification of individualized latent factor can be theoretically grounded. The experimental results indicate the proposed method could achieve superior performance under some conditions.

**Strengths:**

1.	The paper is well structured and easy to follow.

2.	In practice, many individual-specific factors are unobservable, which is a great challenge to accurately learn individual latence. Thus, the work investigates an important problem, from my perspective.

3.	Some assumptions to achieve theoretical guarantees are clearly provided and described.

4.	Codes are provided via GitHub link.

**Weaknesses:**

1.	Lack warranty in some major claims. Though without references, I can understand some reasons described by the authors why only considering discrete representations, since some underlying latence of humans could be discrete such as gender, and continuous representations can be translated into discrete. However, the authors state that the discrete formulation allows for a more nuanced representation of individualized latent, which is lack theoretical and empirical evidence. If that is an assumption, I would also recommend such strong assumption to be clearly formulated and described in main content, given it plays a key role in the methodology.

2.	Design of methodology is not well motivated, and some parts lack details. For example, (a) The motivation of choosing VAE to estimate individual-specific factors, and the motivation of state reconstruction are not clear. The part of feature extraction is not introduced in overview, and the definition of features, and the reason of utilizing feature extraction, are not clear. (b) The proposed method relies on subgrouping of individuals to obtain embedding dictionary, while the part of subgrouping is highly vague in paper and lack thorough motivation and details. For example, (a). There is no reference regarding the motivation of such design, and reasons of choosing the specific subgrouping measure. (b) q is not included in pseudocode. Specific measures to obtain q and how q is determined in experiments are not clear.

3.	Design and details of experimental are not clear, which hinders the evaluation of proposed method and reproducibility. Some important details, such as: for synthetic experiments, the goal of the task, how the synthetic datasets are conducted, definitions of states, actions, and rewards, why only 100 samples are generated, how the unobserved latent L and noise are simulated, as well as the reasons of design such synthetic environment without considering existing broadly used environments (e.g., Adroit, Mujoco), especially human-involved ones. For persuasion for good environment, objectives, states, actions, and rewards are not formally defined. Definition of relevant information from the dialogues using the BERT is not provided.

4.	Gaps exist across motivation, methodology, and experimental design. (a) The major work is motivated by complex human-related scenarios, and it is comprehensive that the paper described a lot of educational and healthcare scenarios in major content. But the experimental environments are mainly simple without well-designed human factors as described in motivation. There is one human-related environment used in experiment, i.e., persuasion for social good in the dialogue system. But details of experiments are missing too much for me to thoroughly evaluate the proposed method. (b) For compared methods, it would be more persuasive if they contain: (i) Population level component with the framework, e.g., learning population level embeddings and reconstructing states, so that the effectiveness of learning individualized factors can be isolated to evaluate; (ii) Different state reconstruction methods such as trajectory transformer, GAN, etc. Otherwise, a clear motivation of current baseline selection and choices of utilized techniques in methodology need to be justified. (c) As what I described in (2), motivations are lack in each step of methodology, hinders a thorough evaluation regarding the proposed method.


Minor:
-	Caption of Figure 3(a): “XX” looks like a placeholder.

-	Limitation should be discussed. A possible limitation is that the individual-specific factor L is assumed to be static over time, while another common fact is that many unobserved factors related to humans can evolve over time (e.g., [1-2]), such as mental status, and affect human behaviors.


[1] Feng, F., Huang, B., Zhang, K., & Magliacane, S. (2022). Factored adaptation for non-stationary reinforcement learning. Advances in Neural Information Processing Systems, 35, 31957-31971.

[2] Mannering, F. L., Shankar, V., & Bhat, C. R. (2016). Unobserved heterogeneity and the statistical analysis of highway accident data. Analytic methods in accident research, 11, 1-16.

**Questions:**

Please see Weaknesses.

---

> ### Author Response · Authors · 2023-11-19
> **Rebuttal by Authors [Part 1]**
>
> **Q1: Lack warranty on why discrete formulation is a more nuanced representation.**
>
> **A1:** Thank you for pointing that out. We are sorry for the confusion and we have removed this part. A new revised version is provided in Section 4.3:
>
> > Group determinacy indicates that for a given task, the individual-specific factors are finite. This is a realistic assumption in numerous practical situations. For example, in hypertension diagnosis, patient groups may be formed based on factors such as age, gender, or medical history. Each group would demonstrate unique health outcomes and patterns, driven by their respective latent individual-specific factors $l_k$.
>
> > we can divide the continuous latent confounder $L$ into small segments and represent each with a discrete value $l_k$.
>
>
> **Q2: The motivation of choosing VAE to estimate individual-specific factors is not clear.**
>
> **A2:** Thank you. Our proposed framework is carefully customized to meet the requirements of the identifiability theorem. According to the definition of identifiability (detailed in Appendix A.3.1), if a learned generative model matches the true model in terms of joint distribution, it would help establish the identifiability of latent individual-specific factors.
>
> This motivates our work to use a generative model and to estimate confounders by fitting the learned distribution to the true observed distribution. Specifically, we use an encoder to infer the low-dimensional latent confounder, and a decoder to mimic the data generation process. This naturally leads us to use the VAE framework. We also found that the performance of the traditional VAE is not sufficient for recovering the latent individual-specific factor, as shown in the ablation study in Section 5.1. Thus, we have developed an individual-specific factor estimation framework to recover the latent factors.
>
> **Q3: The motivation of state reconstruction is not clear.**
>
> **A3:**  According to the identifiability definition, the better the data generation process is learned, the better $\hat{l}_k$ would be recovered. To reconstruct individualized transition processes, we design a conditional decoder to reconstruct the state. The discrepancy between the reconstructed state and the original state provides the reconstruction loss for the optimization of the entire framework.
>
> **Q4: The motivation of feature extraction.**
>
> **A4:**  Sorry for the confusion. "Feature" refers to the output of the encoder before discretization. It can be viewed as a continuous representation of the latent individual-specific factors. To avoid any misunderstanding, we have replaced the original term "feature" with "latent low-dimensional continuous representation."
>
> **Q5: Subgrouping is highly vague in paper and lacks thorough motivation and details.**
>
> **A5:** Regarding the subgroup you mentioned, we guess you are referring to the embedding dictionary in the vector quantization layer.
>
> The concept of the vector quantization layer originates from the paper [1]. his process converts continuous representations into a discrete latent space by identifying the closest vector in a predefined embedding dictionary $E$. The quantized vector $q$, denoted as $\hat{l}$, represents the embedding $e_i$ that is closest to the encoder output $z$, mathematically expressed as $\hat{l}\_k=\arg\min\_{e\_i}\lVert z^k-e\_i\lVert\_2$. This measurement is commonly employed in other related works focused on quantization [2-3].
>
> In the experiment, $\hat{l}_k$ is determined through the process of vector quantization, and the learning process is optimized based on the criterion of minimizing the Euclidean distance between $z$ and $e_i$. We have revised Section 4 and the pseudocode to make this clearer and to avoid any confusion.
>
> > Considering the group determinacy assumption, the estimated latent individual-specific factor should be discrete. We then embed a quantization layer to discretize the continuous latent representation $z^k$ to the latent individual-specific factor $\hat{l}_k$.
>
> > The assignment of a dictionary vector $e_i$ to $z^k$ can be realized by finding the nearest neighbor in the dictionary: $\hat{l}\_k=\arg\min\_{e_i}\lVert z^k-e_i\lVert\_2$, where $\hat{l}\_k$ represents the quantized vector that is the closest embedding $e_i$ to the continuous representation $z^k$.
>
> [1] Van Den Oord A, Vinyals O. Neural discrete representation learning[J]. Advances in neural information processing systems, 2017, 30.
>
> [2] Yu J, Li X, Koh J Y, et al. Vector-quantized image modeling with improved vqgan[J]. arXiv preprint arXiv:2110.04627, 2021.
>
> [3] Gupta A, Mukhopadhyay R, Balachandra S, et al. Towards Generating Ultra-High Resolution Talking-Face Videos with Lip synchronization[C]//Proceedings of the IEEE/CVF Winter Conference on Applications of Computer Vision. 2023: 5209-5218.

---

> ### Author Response · Authors · 2023-11-19
> **Rebuttal by Authors [Part 2]**
>
> **Q6: Design and details of the experiment are not clear in the synthetic experiment.**
>
> **A6:** Thank you for pointing this out. We have updated Section 5 with more experimental details for clarity. Here is what we have added: the task's goal, the method of creating synthetic datasets, the definition of states and actions, and a concise summary and emphasis on how we simulated the unobserved latent factor $L$ and noise in Section 5.1. We did not introduce a reward because it is not essential for evaluating the latent individual-specific factor. Figure 3 is corrected and the placeholder is removed. For additional details, please refer to Appendix D. Specifically, the modifications are as follows:
>
> > We manually created time series data based on the post-nonlinear model and customize different types of latents with the required assumptions satisfied or violated. The data generation process is modeled as $\vec{s}\_t=f\_2(f\_1(\vec{s}\_{t-1},\vec{a}\_{t-1},L),\epsilon\_t)$, where $f\_1$ represents the nonlinear effect, and $f\_2$ denotes the invertible post-nonlinear distortion on $\vec{s}\_t$. The state $\vec{s}\_t\in\mathbb{R}^3$ and actions $\vec{a}\_t\in\mathbb{R}^2$ are initially generated randomly following a uniform distribution, and the noise is modeled as a Gaussian distribution. Three types of $L$ are generated to satisfied or violate our assumptions. Case 1: $L$ is a discrete latent scalar follows the categorical distribution $\text{Cat}(0.1,0.2,0.3,0.4)$. Case 2: $L$ are multiple discrete latent variables follow the categorical distributions $\text{Cat}(0.1,0.9)$, $\text{Cat}(0.2,0.3,0.5)$, $\text{Cat}(0.1,0.2,0.3,0.4)$ respectively. Case 3: $L$ are multiple continuous latent variables follow the Gaussian distribution $\mathcal{N}(0,1)$, uniform distribution ${\rm Uniform}(0,1)$, and exponential distribution ${\rm Exp}(1)$.
>
> **Q7: The reasons why design such synthetic environments without considering existing broadly used environments, especially human-involved ones.**
>
> **A7:** To empirically validate the identifiability results for the individual-specific factor, we need a dataset with a known ground truth that is guaranteed to satisfy the proposed assumptions, including the presence of individual-specific factors. Unfortunately, this condition cannot be directly met by the existing widely-used environments. Therefore, this serves as one of the motivations for using a synthetic dataset for our validation purposes.
>
> Additionally, we can easily modify the synthetic environments to explore what happens when specific assumptions are violated and how it affects the identifiability performance, providing more flexibility to either violate or satisfy our assumptions. Specifically, in Section 5.1, we customized three different types of latent variables involving both continuous/discrete and single/multiple confounders. If we use existing environments, we cannot customize and generate the desired dataset.
>
> It is worth mentioning that we have also modified the Pendulum environment, which is a widely-used RL benchmark, to satisfy the individualized property and generate a population dataset.
>
> **Q8: Why only 100 samples are generated in the synthetic environment.**
>
> **A8:** The key to our sample sufficiency assumption is that the trajectory length should be greater than the number of groups; the number of individuals is less important. In the experiment, we set the number of individuals as 100 to demonstrate the effectiveness of the proposed method. Moreover, to address your concern, we will vary the number of samples to verify the effectiveness of our method. The experiments are still running due to the limited time, and we will update the results in openreview as soon as possible.
>
> **Q9: For persuasion for good environment, objectives, states, actions, and rewards are not formally defined.**
>
> **A9:** Thank you. In the revised version, we have defined the objectives, states, actions, and rewards in the persuasion for good dataset in Section 5.1.
>
> > We further evaluate the effectiveness of the estimation framework on a real-world, open-source dataset, the Persuasion For Good Corpus~\citep{mydataset,wangetal2019persuasion}. It is a collection of dialogues focused on charitable donations, and each dialog involves a persuader trying to convince a persuadee to donate to a charity, where all participants have undergone personality assessments and have corresponding individualized personalities. The state is the persuadee's response, the action is the persuader's utterance, and the reward is the final donation in each dialog. The goal is to make an individualized policy for each persuader to successfully persuade the persuadee to make a charitable donation. To achieve this, we first need to identify the latent personality of each individual from the offline dataset.

---

> ### Author Response · Authors · 2023-11-19
> **Rebuttal by Authors [Part 3]**
>
> **Q10: The definition of relevant information from the dialogues using the BERT is not provided. The details of experiments are missing too much to thoroughly evaluate the proposed method.**
>
> **A10:** In the revised version, we have added more details on the Persuasion for Good dataset. The information on how we extract relevant information from the dialogues using BERT has been included in Appendix D to save space in the main paper.
>
> > In our experiment, we use a pre-trained BERT model to convert each dialogue into vectors. The process starts with tokenization, breaking words into smaller units. These tokens are then input into BERT, which provides contextual embeddings for each token, considering the entire dialogue context.
>
>
> **Q11: The experimental environments are mainly simple without well-designed human factors as described in motivation.**
>
> **A11:** We are sorry about it. We tried our best to find more experimental environments involving human factors. However, we failed to find any other datasets except the Persuasion For Good Corpus dataset.
>
>
> **Q12: Why do not use different state reconstruction methods such as trajectory transformer, GAN, etc.**
>
> **A12:** Thank you for the question. The trajectory transformer's involves discretizing continuous states and actions. We have not identified a need for discretization in state generation.
>
> Regarding GANs, their training process is known to be less stable than VAEs. Achieving stability with GANs often necessitates more extensive hyperparameter tuning. Therefore we use VAE-based method in our original paper.
>
>
> **Q13: A clear motivation of current baseline selection and choices of utilized techniques in methodology need to be justified.**
>
> **A13:** Thanks for your great advice. We have restructured Section 5 and clearly outlined the motivation behind our current baseline choices:
>
> > (1) We experimented with other sequential generative model baseline, namely the disentangled sequential autoencoder. It separates the latent representations into static and dynamic parts, in contrast to our method which focuses on the global influence of $L$.
>
> > (2) The VQ-VAE serves as the base model for ablation. Model variants are built upon it to disentangle the contributions of different modules.
>
> > (3) Adaptation to continuous latent variables allows us to evaluate the flexibility of our method and to see if the continuous latent assumption distorts identifiability.
>
> Moreover, we have emphasized the choices of utilized techniques in Section 4, which have also been addressed in A2-A5, and A15.

---

> ### Author Response · Authors · 2023-11-19
> **Rebuttal by Authors [Part 4]**
>
> **Q14: Compare with population level component with the framework**
>
> **A14:** Thank you for your insightful suggestion. We have included the population level component as baselines in our experiment. The experiments are still running due to the limited time, and we will update the results in openreview as soon as possible.
>
>
> **Q15: Motivations are lacking in each step of methodology.**
>
> **A15:** Thanks for your feedback. We have thoroughly revised the manuscript to clearly explain the function and importance of each component. Specifically, we have improved Section 4 by adding the following:
>
> 1) An overview paragraph that outlines the design of our methodology and provides a brief summary of the overall framework.
>
> > Our proposed framework is carefully customized to meet the requirements of the identifiability theorem. According to the identifiability definition A.3, latent confounders are identifiable if observational equivalence can consistently lead to the identifiability of the latent variables. This motivates our work to utilize a generative model and achieve confounder estimation by fitting the learned distribution to the true observed distribution. Traditional VAE encodes the input data into a continuous latent space using probabilistic encoders and then reconstructs the input from this space using decoders. However, this approach is not suitable for our problem setting. Specifically, given the offline trajectory of the population, we have customized the VAE framework and developed an individual-specific factor estimation framework to recover the latent factors.
>
> > First, to obtain latent representations in a reduced-dimensional space, we use an encoder with the entire sequence $\vec{s}^k\_{0:T}$ of each individual $k$ as input and the latent continuous representation $z^k$ as output. Considering the group determinacy assumption, the estimated latent individual-specific factor should be discrete. We then incorporate a quantization layer to discretize the continuous latent representation $z^k$ into the latent individual-specific factor $\hat{l}\_k$. Intuitively, the better $\hat{l}\_k$ is recovered, the more effectively the data generation process can be learned. To reconstruct individualized transition processes, we design a conditional decoder that takes the conditions $(\vec{s}^k\_{t-1},\vec{a}^k\_{t-1})$, the estimated latent individual-specific factor $\hat{l}\_k$, and the noise term $\hat{\nu}\_k$ as inputs, producing $\hat{\vec{s}}^k\_t$ as the output. The noise is estimated based on $\hat{l}\_k$ to account for the stochasticity of the state transition processes. Afterward, the optimized quantized encoder is employed to augment the original buffer and facilitate downstream individualized policy learning.
>
> 2) A straightforward explanation of the functions of each component, including its inputs/outputs and essential mathematical formulations. Specifically,
>
> - In the quantized encoder module, we have added the motivation for introducing the quantization layer, along with corresponding update functions. The changes are as follows:
>
> > The group determinacy assumption implies the existence of the latent individual-specific factor $L$. Since $L$ denotes the time-invariant latent individual-specific factors that influence each state in the transition process, we initially use an encoder to capture the continuous representation $z^k$ from $\vec{s}^k\_{0:T}$, and employ a quantization layer to delineate unique and distinct groups as $\hat{l}^k$.
>
> > The corresponding update functions are $o^k\_t=\text{Conv1D}(\vec{s}^k\_{t:t+H})$, $h^k\_t,c^k\_t=\text{LSTM}(h^k\_{t-1},c^k\_{t-1},\vec{s}^k\_t;\theta)$, where $o^k\_t$, $h^k\_t$ and $c^k\_t$ are the feature map, hidden state and cell state at time $t$, and $\theta$ denotes all the parameters of the LSTM.
>
> - In the noise estimation module, we have provided the motivation for including this module and explained how it applies within our framework.
>
> > To account for the stochastic nature of the individualized transition processes, a noise estimator is introduced with $\hat{l}^k$ as input and $\hat{\nu}_k$ as output.
>
> > The motivations for introducing a noise layer are 1) to mimic the probabilistic property of the individualized state transition in Equation 1, and 2) to compensate for the loss of stochasticity due to the deterministic nature of the quantization operation.
>
> > Our posterior $ q(\hat{l}\_k=\kappa|\vec{s}^k_{0:T}) = \\{1, \text{for } \kappa=\arg\min\_j \lVert z^k-e\_j\lVert\_2; 0\text{ otherwise.}\\}$ is a categorical distribution and becomes deterministic after the quantization operation. By adding noise, we reintroduce variability that allows the model to better varying data, thus improving the model's ability to generalize and increase robustness in different scenarios.

---

> ### Author Response · Authors · 2023-11-19
> **Rebuttal by Authors [Part 5]**
>
> **Q16: A possible limitation is that the individual-specific factor L is assumed to be static over time, while another common fact is that many unobserved factors related to humans can evolve over time.**
>
> **A16:** Regarding the related works that assume unobserved factors can evolve over time, we would like to emphasize that the latent latent individual-specific factor we consider in this work are allowed to influence each state in the decision process and vary across individuals, which is very different from the existing works [1-2].
>
> Specifically, existing works [1-2] usually focus on latent variables that are time-varying. When they are time-varying, they can benefit from many recent advances in nonlinear ICA to achieve strong identifiability results [3-4]. However, the identifiability of time-invariant latent confounders, though not well-studied, has numerous applications. For example, in healthcare, two patients diagnosed with hypertension may have different health outcomes (state $s_t$) after receiving the same treatment (action $a_t$), influenced by the time-invariant latent confounders ($L$) such as gene variations, birth conditions, or genetic predispositions. If we could observe these predispositions, it would greatly assist physicians in predicting how patients will respond to specific treatments and facilitate personalized healthcare. However, in most cases, these individual-specific factors remain latent and are difficult to diagnose. Therefore, the theoretical identification of such traits and the development of an estimation framework to extract them from observational data is of immense practical value. This formulation and understanding significantly influence individualized decision-making in downstream tasks.
>
> [3] Hyvarinen A, Morioka H. Unsupervised feature extraction by time-contrastive learning and nonlinear ica[J]. Advances in neural information processing systems, 2016, 29.
>
> [4] Yao W, Sun Y, Ho A, et al. Learning temporally causal latent processes from general temporal data[J]. arXiv preprint arXiv:2110.05428, 2021.

---

> ### Author Response · Authors · 2023-11-22
> **Follow-Up Responses by Authors (Experimental Results)**
>
> Dear Reviewer y8Yq,
>
> Thank you once again for your insightful suggestions to improve our paper. We now have the updated experiment results for **Q8** and **Q14**, and the details are as follows.
>
> **Follow-Up Q8: Vary the number of samples to verify the performance.**
>
> **Follow-Up A8:** We varied the number of samples to $\\{100,150,200,300,500,800,1000\\}$ and compared the PCC trajectories. The result shows that our method can achieve consistently good recovery performance under different numbers of individuals.
>
> The additional experiment results have been added in Appendix D.3. Specifically,
>
> > In the main paper, we set the number of individuals as 100 to demonstrate the effectiveness of our proposed method. Here, we will vary the number of samples to further verify this effectiveness.
> The data generation process is consistent with Case 1, except that we change the number of samples to $\\{100,150,200,300,500,800,1000\\}$. The comparison results shown in Figure 5 indicate that our method can achieve consistently good recovery performance under different numbers of individuals. This further confirms that the identifiability of our framework is guaranteed by the mathematical relationship between the trajectory length and the number of groups, which is constrained by the sample sufficiency assumption; the number of individuals is of lesser importance.
>
>
> **Follow-Up Q14: Compare our method with population level component with the framework.**
>
> **Follow-Up A14:** We have included the Population-Level Component (PLC) as one baseline in our experiment. The PCL takes the population-level data as input, while our method uses individual-level trajectories. Consequently, it is unable to derive individual-specific factors.
>
> The result shows that our method outperforms the PCL, suggesting that population-level embeddings are insufficient for the effective reconstruction of individualized transition processes. This further confirms the effectiveness of learning individual-specific factors.
>
> We have revised the description of the baselines and comparison results in Section 5. Specifically,
>
> > (1) Disentangled Sequential Autoencoder (DSA), which separates the latent representations into static and dynamic parts instead of considering the global influence of $L$.
> > (2) Population-Level Component (PLC), which learns the population-level embeddings instead of focusing on individualized factors.
> > (3) VQ-VAE serves as the base model for ablation. Model variants are built upon it to disentangle the contributions of different modules.
> > (4) Adaptation to continuous latent variables, which allows us to evaluate the flexibility of our method and to assess if the continuous latent assumption distorts identifiability.
>
> > The comparison result in Fig.3(a) shows that our method outperforms the baselines. Specifically, DSA reconstructs the entire time series but overlooks individualized transition processes, resulting in compromised identifiability and deteriorating recovery performance over time. PCL lacks the individual-specific factor to delineate distinct groups thus cannot achieve identifiability.
>
> We hope that your comments are properly addressed, and look forward to receiving your feedback at your earliest convenience. We are more than happy to respond to it.
>
> Sincerely,
>
> Authors

---

> > ### Comment · Reviewer_y8Yq · 2023-11-22
> > **Response by reviewer y8Yq**
> >
> > I would like to thank the authors for the response. And I really appreciate the authors’ time and efforts in adding relevant baselines, varying sample sizes, and updating manuscript in such a short time.
> >
> > I am satisfied with some responses. For the ones I am still concerning about/unsure, please see my comments below.
> >
> > (Minor) Definition A.3 is mentioned as a motivation in methodology, while it is put in Appendix, hindering the readability. The major content should be self-contained. I can understand that can be due to the space limitation, but the authors are recommended to get avoid of that in future versions.
> >
> > A3,  A4, A5: need proper references to support the statements of intuitions in manuscript, if possible. Also, it looks like some formatting bugs exist in the response (i.e., A5). Anyway, I can get the major points of the response.
> >
> > A7. I didn’t get the exact reasons for not considering some popular benchmarks (e.g., Adroit, Mujoco), with ground truth can be obtained from them. If those don’t fit the authors’ assumptions, please clearly state. I agree Pendulum is also broadly used, but simple. The reason I was asking that is I’m concerned with the generalizability and robustness of the work, especially in varied tasks and complex environments, since real-world human-related environments can be much more complex, and it would be more persuasive to conduct more thorough evaluations on more complex environments.
> >
> > A11. Maybe Adroit ‘human’ datasets can be considered.
> >
> > A16. In my original response, my major point is that limitation should be properly and explicitly described in the main content (which is omitted in the authors’ response). And I was trying to point out a possible one from my perspective. Please consider adding limitations of your work in the manuscript, for a more thorough comprehension.
> >
> > Anyway, I see the authors’ efforts in improving their work. I’d like to update my score to 5 at this moment. Thanks.

---

> > > ### Author Response · Authors · 2023-11-23
> > > **Responses by Authors to Follow-Up Questions**
> > >
> > > Dear Reviewer y8Yq,
> > >
> > > Thank you for recognizing our revisions and providing constructive feedback. We are happy to address your concerns in detail below.
> > >
> > > **Follow-Up Q1: Relocate Definition A.3 to the major content.**
> > >
> > > **Follow-Up A1:** Thank you for your valuable advice. In the revision, we have relocated Definition A.3 to Section 4, a newly created section related to theorems, to enhance readability and ensure the paper is self-contained.
> > >
> > >
> > > **Follow-Up Q2: A3, A4, A5 need proper references to support the statements of intuitions in manuscript.**
> > >
> > > **Follow-Up A2:** Thank you for your advice, and we apologize for the formatting issues in A5. We have now included detailed references in each response, which you may check in the following response (we are unable to edit the original response due to characteristic limitations). Specifically,
> > >
> > > For A3, we added reference [1] to present similar ideas about introducing extra conditions to the decoder for state reconstruction.
> > >
> > > For A4, we included references [2-3] to support the widespread use of encoders in mapping high-dimensional data into a low-dimensional feature space.
> > >
> > > For A5, we corrected the formatting issues and added reference [4] to discuss similar ideas on using an embedding dictionary for vector quantization.
> > >
> > > [1] Kittichokechai K, Oechtering T J, Skoglund M. Coding with action-dependent side information and additional reconstruction requirements[J]. IEEE Transactions on Information Theory, 2015, 61(11): 6355-6367.
> > >
> > > [2] Ma X, Lin Y, Nie Z, et al. Structural damage identification based on unsupervised feature-extraction via Variational Auto-encoder[J]. Measurement, 2020, 160: 107811.
> > >
> > > [3] Yao R, Liu C, Zhang L, et al. Unsupervised anomaly detection using variational auto-encoder based feature extraction[C]//2019 IEEE International Conference on Prognostics and Health Management (ICPHM). IEEE, 2019: 1-7.
> > >
> > > [4] Marimont S N, Tarroni G. Anomaly detection through latent space restoration using vector quantized variational autoencoders[C]//2021 IEEE 18th International Symposium on Biomedical Imaging (ISBI). IEEE, 2021: 1764-1767.
> > >
> > > **Follow-Up Q3 (related to A7 & A11): Concerns about the generalizability and robustness of the work, especially in varied tasks and complex environments.**
> > >
> > > **Follow-Up A3:** We agree with your suggestion to experiment in more complex reinforcement learning environments to further validate our algorithm's effectiveness. We are particularly grateful for your recommendation to use the Adroit-Human datasets. However, given the approaching rebuttal deadline and the extensive training time required in an Adroit or Mujoco environment, we are constrained in this regard.
> > >
> > > Thank you for your understanding and valuable feedback. We will update new experiment results in the final version of our paper.
> > >
> > >
> > > **Follow-Up Q4 (related to A16): Limitations should be properly and explicitly described in the main content.**
> > >
> > > **Follow-Up A4:** We apologize for not mentioning the revised limitations in our previous response. We highly value your feedback and have added a detailed discussion in Conclusion in our revised manuscript. Please kindly let us know if this addresses your concern.
> > >
> > > We mention two limitations that arise from our assumptions: (1) no instantaneous causal influence is allowed within $\vec{s}\_t$, and (2) the latent confounder is discrete. Extending our identifiability theories and framework to account for such properties is a direction for our future work. Specifically, we have added a discussion of these limitations in the main content as follows.
> > >
> > > > The main limitations of this work lie in our two main assumptions: (1) there is no instantaneous causal influence within $\vec{s}\_t$, and (2) the latent confounder is discrete. Instantaneous relationships, if present, may distort the identifiability of the latent state-transition process, although their impact can be moderated by adjusting the temporal resolution of the data. Although our theoretical framework does not extend to scenarios with evolving continuous latents, our empirical results suggest potential for adapting our approach to a broader range of scenarios. Extending our identifiability theories and framework to account for such properties is our future direction.

---

> > > ### Author Response · Authors · 2023-11-23
> > > **Updated Responses by Authors (Regarding A3, A4, A5)**
> > >
> > > **Q3: The motivation of state reconstruction is not clear.**
> > >
> > > **Updated A3:**  According to the identifiability definition, the better the data generation process is learned, the better $\hat{l}_k$ would be recovered. To reconstruct individualized transition processes, we design a conditional decoder for state reconstruction. Introducing extra conditions to the decoder can facilitate this process, and related work shares similar ideas [1]. The discrepancy between the reconstructed state and the original state provides the reconstruction loss for the optimization of the entire framework.
> > >
> > > [1] Kittichokechai K, Oechtering T J, Skoglund M. Coding with action-dependent side information and additional reconstruction requirements[J]. IEEE Transactions on Information Theory, 2015, 61(11): 6355-6367.
> > >
> > > **Q4: The motivation of feature extraction.**
> > >
> > > **Updated A4:**  Sorry for the confusion. "Feature" refers to the output of the encoder before discretization. It can be viewed as a continuous representation of the latent individual-specific factors. Generally, encoders are widely used to map high-dimensional data into a low-dimensional feature space [1-2]. To avoid any misunderstanding, we have replaced the original term "feature" with "latent low-dimensional continuous representation."
> > >
> > > [1] Ma X, Lin Y, Nie Z, et al. Structural damage identification based on unsupervised feature-extraction via Variational Auto-encoder[J]. Measurement, 2020, 160: 107811.
> > >
> > > [2] Yao R, Liu C, Zhang L, et al. Unsupervised anomaly detection using variational auto-encoder based feature extraction[C]//2019 IEEE International Conference on Prognostics and Health Management (ICPHM). IEEE, 2019: 1-7.
> > >
> > > **Q5: Subgrouping is highly vague in paper and lacks thorough motivation and details.**
> > >
> > > **Updated A5:** Regarding the subgroup you mentioned, we guess you are referring to the embedding dictionary in the vector quantization layer.
> > >
> > > The concept of the vector quantization layer originates from the paper [1] and related work shares the similar ideas on using an embedding dictionary for vector quantization [2]. This process converts continuous representations into a discrete latent space by identifying the closest vector in a predefined embedding dictionary $E$. The quantized vector $q$, denoted as $\hat{l}$, represents the embedding $e_i$ that is closest to the encoder output $z$, mathematically expressed as $\hat{l}\_k=\arg\min\_{e\_i}\lVert z^k-e\_i\lVert\_2$. This measurement is commonly employed in other related works focused on quantization [3-4].
> > >
> > > In the experiment, $\hat{l}_k$ is determined through the process of vector quantization, and the learning process is optimized based on the criterion of minimizing the Euclidean distance between $z$ and $e_i$. We have revised Section 4 and the pseudocode to make this clearer and to avoid any confusion.
> > >
> > > > Considering the group determinacy assumption, the estimated latent individual-specific factor should be discrete. We then embed a quantization layer to discretize the continuous latent representation $z^k$ to the latent individual-specific factor $\hat{l}_k$.
> > >
> > > > The assignment of a dictionary vector $e_i$ to $z^k$ can be realized by finding the nearest neighbor in the dictionary: $\hat{l}\_k=\arg\min\_{e_i}\lVert z^k-e_i\lVert\_2$, where $\hat{l}\_k$ represents the quantized vector that is the closest embedding $e_i$ to the continuous representation $z^k$.
> > >
> > > [1] Van Den Oord A, Vinyals O. Neural discrete representation learning[J]. Advances in neural information processing systems, 2017, 30.
> > >
> > > [2] Marimont S N, Tarroni G. Anomaly detection through latent space restoration using vector quantized variational autoencoders[C]//2021 IEEE 18th International Symposium on Biomedical Imaging (ISBI). IEEE, 2021: 1764-1767.
> > >
> > > [3] Yu J, Li X, Koh J Y, et al. Vector-quantized image modeling with improved vqgan[J]. arXiv preprint arXiv:2110.04627, 2021.
> > >
> > > [4] Gupta A, Mukhopadhyay R, Balachandra S, et al. Towards Generating Ultra-High Resolution Talking-Face Videos with Lip synchronization[C]//Proceedings of the IEEE/CVF Winter Conference on Applications of Computer Vision. 2023: 5209-5218.

---

### Official Review · Reviewer_rs8u · 2023-11-06

**Soundness:** 2 fair
**Presentation:** 2 fair
**Contribution:** 1 poor
**Rating:** 3
**Confidence:** 3

**Summary:**

The authors propose an approach for individualized RL where policies can be tailored to individual people/agents. Their approach is built on the idea of learning latent state-transition processes using a VAE. They demonstrate the empirical performance of their approach on a series of synthetic and real-world envirnoments.

**Strengths:**

- The authors motivate the work with a wide range of applications, from medical applications to educational applications.
- Some of the diagrams demonstrating the workflow/algorithm are quite nice, although the font is hard to read in some places.

**Weaknesses:**

- The communication is unclear in some parts. It would perhaps be helpful to motivate the work with a single running example rather than jumping around between applications.
- The font size in the plots is too small to read.
- I'm not sure that pendulum control counts as a real-world application.
- The contribution of the paper is quite small. Much of the contribution is simply incorporating a VAE into an RL system.

**Questions:**

If you had to choose one application where you think your method would truly shine, what would it be?

---

> ### Author Response · Authors · 2023-11-19
> **Rebuttal by Authors [Part 1]**
>
> **Q1: The communication is unclear in some parts. It would perhaps be helpful to motivate the work with a single running example rather than jumping around between applications.**
>
> **A1:** Thanks for your great advice. Following your suggestions, we have incorporated the healthcare example mentioned in the Introduction and included it as a running example. The changes are as follows:
>
> (1) In Section 3, we introduce the concepts of state, action, reward, and latent individual-specific factors in the case of hypertension diagnosis.
>
> > For example, in the context of hypertension diagnosis. The patient's health status at each time can be represented as the state $\vec{s}\_t$, the individualized treatment can be viewed as the action $\vec{a}\_t$, and the blood pressure level is the reward $r_t$. The latent confounder $l$ represents the unobserved genetic predispositions that are unique to each patient and have a significant impact on their long-term health, such as gene variations. In this case, the health status $\vec{s}\_t$ at time $t$ is determined by the previous status $\vec{s}\_{t-1}$, previous treatment $\vec{a}\_{t-1}$, together with the intrinsic gene variations. Such implicit associations are indicated by the red dashed lines. If we could observe these predispositions, it would greatly help physicians to predict how patients will respond to specific treatments, enabling personalized care. In most cases, however, these individualized factors remain latent and difficult to diagnose. Therefore, the theoretical identification of such traits and the development of an estimation framework to extract them from observed data is of great practical value.
>
> (2) In Section 4.3, we use hypertension diagnosis to provide an intuitive explanation on the assumptions.
>
> > Group determinacy indicates that for a given task, the individual-specific factors are finite. This is a realistic assumption in numerous practical situations. For instance in hypertension diagnosed. Even if they receive the same treatment they might experience different health outcomes. These differences can be attributed to latent factors like their medical history, which are essentially finite records of past medical events and treatments. In this case, we can group patients based on their history. Each group would demonstrate unique health outcomes and patterns, driven by their respective individual-specific factors $l_k$.
>
> (3) In Appendix A.2, we use the example of hypertension diagnosis to provide intuitive explanations for each assumption and discuss their relevance to real-world applications.
>
> > As for the applicability of group determinacy assumption: The idea of group determinacy is important in real-world applications, such as diagnosing hypertension in healthcare.  Here, $\vec{s}^k$ represents the health condition of a patient, $\vec{a}^k$ is their tailored treatment, and the function $f^k$ decides the unique health trajectory for each patient group. The latent individual-specific factor $l_k$ influences how a patient's health changes over time. It can be based on factors such as age, gender, or medical history that help to group patients logically. For example, one group might be younger people without hypertension, and another group might be older people with different medical histories. Each group has its own set of health outcomes and patterns, influenced by $l_k$ and guided by $f^k$.
>
> > As for the applicability of sample sufficiency assumption: For instance, in diagnosing hypertension, transition samples refer to the tuple of patients' past and present health states along with their treatments $\{(\vec{s}\_{t-1},\vec{a}\_{t-1}, \vec{s}\_t)\}^T_{t=1}$.
>
> > As for the applicability of sequential dependency assumption: In hypertension management, for example, this assumption implies that a patient's current health status and treatment decisions directly affect their next health state and immediate health outcomes, allowing for more precise and effective treatment planning. It also allows healthcare providers to analyze patient data, such as blood pressure readings, at a specific point in time without the influence of unrelated factors. This independence is critical for accurate diagnosis and treatment of hypertension, as it enables individualized patient care by focusing on the specific factors relevant to each individual's health journey.
>
> **Q2: The font size in the plots is too small to read.**
>
> **A2:** Thank you for pointing this out. We have increased the font size and clarity of all figures to improve readability in the revised manuscript, and Figure 2 has been redrawn to better illustrate the framework's structure.

---

> ### Author Response · Authors · 2023-11-19
> **Rebuttal by Authors [Part 2]**
>
> **Q3: I'm not sure that pendulum control counts as a real-world application.**
>
> **A3:** Thank you for pointing this out. We have changed the section name to 'Evaluation on Policy Learning Framework' to avoid misunderstanding.
>
> **Q4: The contribution of the paper is quite small. Much of the contribution is simply incorporating a VAE into an RL system.**
>
> **A4:** Thanks. We would like to emphasize that our contributions go far beyond the integration of a VAE into a RL system.
>
> To the best of our knowledge, none of the existing works consider a time-invariant latent confounder that influences each state in the transition process. We are the first to model such processes and provide an identifiability result.
>
> Specifically, existing works [1-3] usually focus on latent variables that are time-varying. When they are time-varying, they can benefit from many recent advances in nonlinear ICA to achieve strong identifiability results [4-5]. However, the identifiability of time-invariant latent confounders, though not well-studied, has numerous applications. For example, in healthcare, time-invariant latent confounders could be gene variations, birth conditions, or genetic predispositions, which influence the state transition process of health status.
>
>
> In this work, we aim to bridge this gap. We seek to answer the important question of which types of transition processes can have a theoretical identifiability guarantee. We demonstrate that when the individual-specific factors are finite, our method ensures the identifiability of the entire latent state-transition process, even when the transition processes are nonlinear. This establishes novel theoretical insights for learning state-transition processes with latent individual-specific factors.
>
> Furthermore, motivated by the identifiability results, we have designed a practical generative framework, aligning with our theoretical assumptions, **which is very different from VAE**. Specifically,
>
> 1) To satisfy the group determinacy assumption, which assumes there is a finite number of L, we utilize a quantized encoder where the embedding dictionary contains only a finite number of L.
>
> 2) To satisfy the sequential dependency assumption, which assumes the generative process where $s_t$ is determined by $s_{t-1}$, $a_{t-1}$, L and noise, we model the noise with an MLP layer. Then, a conditional encoder, along with a noise layer, is designed to model the data generation process.
>
> To enable readers to quickly grasp the essence of our contribution, we have revised the Introduction as follows:
>
> > We propose the Individualized Markov Decision Processes (iMDPs) framework, a novel approach that integrates individual-specific factors into decision-making processes. We model these factors as latent confounders. We allow them to influence each state in the decision process and to vary across different individuals. This framework has many real-world applications.
>
> > Our framework has theoretical guarantees. We show that when individual-specific factors are finite, our method ensures the identifiability of whole latent state-transition processes, even when the transition processes are nonlinear. This establishes novel theoretical insights for learning state-transition processes with latent individual-specific factors. Additionally, for scenarios with non-finite individual-specific factors, we show that categorizing them into finite groups has little impact on the empirical results.
>
> To the best of our knowledge, this is the first work to provide a theoretical guarantee for the identification of latent individual-specific factors from observed data.
>
> > We propose a practical generative-based method that can effectively infer latent individual-specific factors. Empirical results on both synthetic and real-world datasets demonstrate the method's effectiveness not only in inferring these factors but also in learning individualized policies.
>
> [1] Zhang A, Lyle C, Sodhani S, et al. Invariant causal prediction for block mdps[C]//International Conference on Machine Learning. PMLR, 2020: 11214-11224.
>
> [2] Delgrange, Florent, Ann Nowe, and Guillermo A. Pérez. "Wasserstein Auto-encoded MDPs: Formal Verification of Efficiently Distilled RL Policies with Many-sided Guarantees." arXiv preprint arXiv:2303.12558 (2023).
>
> [3] Guo, Zhaohan Daniel, et al. "Bootstrap latent-predictive representations for multitask reinforcement learning." International Conference on Machine Learning. PMLR, 2020.
>
> [4] Hyvarinen A, Morioka H. Unsupervised feature extraction by time-contrastive learning and nonlinear ica[J]. Advances in neural information processing systems, 2016, 29.
>
> [5] Yao W, Sun Y, Ho A, et al. Learning temporally causal latent processes from general temporal data[J]. arXiv preprint arXiv:2110.05428, 2021.

---

> ### Author Response · Authors · 2023-11-19
> **Rebuttal by Authors [Part 3]**
>
> **Q5: If you had to choose one application where you think your method would truly shine, what would it be.**
>
> **A5:** Thank you for the question. Our work should have many applications in various fields. It is really hard to choose. We would like to provide some examples in healthcare. Both works emphasize the importance of individualized treatment. Specifically,
>
> [1] considers the individualized treatment protocol for treatment-naive BRVO patients with macular edema, where the state can be viewed as visual and health conditions, the action can be viewed as aflibercept injections, rewards can be viewed as visual acuity and central macular thickness improvements, and individual-specific factors can be viewed as the gene factors that influence the patients' responses to anti-VEGF treatment.
>
> [2] considers the management of Cushing's disease, where the state can be viewed as the patient's health condition; actions can be viewed as treatments such as surgery and medical therapy; rewards can be viewed as improvements in health and reductions in hypercortisolism-related issues, and the individual-specific factor can be viewed as the gene factors that influence the clinical and biochemical responses.
>
> > ...some patients respond well to treatment whereas others show poor or no treatment response. For this reason, authors have suggested that treatment protocols should be **individualized**.
>
> [1] Noma H, Yasuda K, Narimatsu A, et al. New individualized aflibercept treatment protocol for branch retinal vein occlusion with macular edema[J]. Scientific Reports, 2023, 13(1): 1536.
>
> > patient management should be **individualized** and lifelong follow-up is required...
>
> [2] Fleseriu M, Varlamov E V, Hinojosa-Amaya J M, et al. An individualized approach to the management of Cushing disease[J]. Nature Reviews Endocrinology, 2023, 19(10): 581-599.
>
> However, there is a fundamental difference between their work and ours: while they do not provide a guaranteed method for identifying individual-specific factors in state transition processes, our research addresses this critical gap. We offer an innovative approach that reliably identifies individual-specific factors from observed data.

---

> ### Author Response · Authors · 2023-11-22
> **Could you please let us know whether our responses and updated submission properly addressed your concern?**
>
> Dear Reviewer rs8u,
>
> Thank you once again for your valuable time reviewing our paper and for your helpful suggestions. We have tried our best to address your concerns in the response and revision. Due to the limited time for rebuttal discussion, we look forward to your feedback at your earliest convenience and the opportunity to respond to it.
>
> Sincerely,
>
> Authors

---

> ### Author Response · Authors · 2023-11-23
> **Rolling discussion coming to an end – thank you and awaiting your feedback**
>
> Dear Reviewer rs8u,
>
> We express our sincere gratitude for the time you've taken to review our paper. Your suggestion regarding the running example has significantly contributed to improving the quality of our paper. We have made detailed revisions to the manuscript and addressed your concerns in our response. We hope that our answers have resolved any concerns. Your feedback is valuable to us, and we would greatly appreciate any further response.
>
> Many thanks,
>
> Authors

---

### Official Review · Reviewer_2bUe · 2023-11-08

**Soundness:** 2 fair
**Presentation:** 3 good
**Contribution:** 2 fair
**Rating:** 6
**Confidence:** 3

**Summary:**

This paper proposes a method for identifying latent state-transition processes for personalized reinforcement learning. The authors introduce a practical approach capable of effectively learning these processes from observed state-action trajectories, supported by theoretical guarantees. They demonstrate the efficacy of their method using both synthetic and real-world datasets, showing its potential to facilitate the learning of tailored RL policies. The paper hilights the significance of comprehending personalized state-transition dynamics to devise individual-specific policies in reinforcement learning.

===

I appreciate the author's clarification. I have increased my score from 5 to 6, given the significance of this research. I hope the later revision can improve the writing quality and acknowledge these limitations, which would be valuable for future work.

**Strengths:**

- The paper is basically well-written and contains no significant grammatical errors
- The proposed method appears to be simple yet effective

**Weaknesses:**

- The novelty is somewhat overclaimed.

There are many works, such as [1-5], on learning latent/factored MDPs, which are very similar to the so-called individualized MDPs. However, the paper does not clearly clarify the difference and contribution.

[1] Kearns, Michael, and Daphne Koller. "Efficient reinforcement learning in factored MDPs." *IJCAI*. Vol. 16. 1999.

[2] Zhang, Xuezhou, et al. "Efficient reinforcement learning in block MDPs: A model-free representation learning approach." *International Conference on Machine Learning*. PMLR, 2022.

[3] Feng, Fan, et al. "Factored adaptation for non-stationary reinforcement learning." *Advances in Neural Information Processing Systems* 35 (2022): 31957-31971.

[4] Guo, Zhaohan Daniel, et al. "Bootstrap latent-predictive representations for multitask reinforcement learning." *International Conference on Machine Learning*. PMLR, 2020.

[5] Delgrange, Florent, Ann Nowe, and Guillermo A. Pérez. "Wasserstein Auto-encoded MDPs: Formal Verification of Efficiently Distilled RL Policies with Many-sided Guarantees." *arXiv preprint arXiv:2303.12558* (2023).]

- There are other possible solutions

This paper seems to claim that there is no prior art in this field; however, the reviewer believes that MARL (Multi-Agent Reinforcement Learning) methods and latent MDP (Markov Decision Process) methods could also be applied to the setting described in this paper.

- The theoretical result requires unrealistic assumptions.

The group determinacy, sample sufficiency, and sequential dependency assumptions are too strong to be realistic. Thus, the obtained theoretical results, though seemingly novel, have no significant implications. The reviewer cannot derive insights from this theorem.


Suggestions:

- The symbol $L$ should depend on $k$ in Equation (1).
- To improve readability, background information about VAE should be provided before applying (VQ-)VAE in Section 4.
- The DDPG method is considered outdated. To show the significance and superiority
- The font size in Figure 3 is too small to be readable.

**Questions:**

Some terminologies are confusing:

- What is the difference between individualized MDPs, partially observed MDPs and multi-agent MDPs?
- What exactly does the identifiability property of the individualized latent factor in Theorem 1 mean?
- The optimal policy appears to be non-stationary in individualized MDPs. How can this difficulty be addressed?

---

> ### Author Response · Authors · 2023-11-19
> **Rebuttal by Authors [Part 1]**
>
> **Q1: As there are many similar works [1-5], the novelty is somewhat overclaimed.**
>
> **A1:** We apologize for the confusion. Our method is fundamentally distinct from existing approaches [1-5], as it is based on a different latent transition process. Both our theoretical results and practical methods are grounded in this unique aspect of the latent transition process, making our approach different from existing methods.
>
> **Here are some key differences in the latent transition process.**
>
> - In our work, we consider a latent confounder (individual-specific factor) that influences each state in the state transition process, and the proposed individualized transition process is motivated by numerous applications. We provide theoretical results that when individual-specific factors are finite, our method ensures the identifiability of the entire latent state-transition process, even in cases of nonlinear transitions. This establishes novel theoretical insights for learning state-transition processes with latent individual-specific factors.
>
> - However, Factored MDP [1] and Factored Non-stationary MDP [3] assume there are **no unobserved confounders** in the state transitions. Block MDP [2], POMDP [4], and Latent MDP [5] consider **latent states/spaces, but such latent factors do not influence each state in the transition process**. The aforementioned settings differ significantly from our work.
>
> **Here we would like to provide more details regarding on the difference.**
>
> **To the best of our knowledge, none of the existing works consider a time-invariant latent confounder that influences each state in the transition process. We are the first to model such processes and provide an identifiability result.**
>
> Specifically, existing works [1-3] usually focus on latent variables that are time-varying. When they are time-varying, they can benefit from many recent advances in nonlinear ICA to achieve strong identifiability results [4-5]. However, the identifiability of time-invariant latent confounders, though not well-studied, has numerous applications.
>
> (For example, in healthcare, two patients diagnosed with hypertension may have different health outcomes (state $s_t$) after receiving the same treatment (action $a_t$), influenced by the time-invariant latent confounders ($L$) such as gene variations, birth conditions, or genetic predispositions. If we could observe these predispositions, it would greatly assist physicians in predicting how patients will respond to specific treatments and facilitate personalized healthcare. However, in most cases, these individual-specific factors remain latent and are difficult to diagnose.)
>
> The theoretical identification of such latent confounders and the development of an estimation method to extract them from observational data is of immense practical value. Understanding such a transition process accelerates the development of individualized machine learning.
>
> In this work, we provide a theoretical understanding of such a transition process.We demonstrate that when individual-specific factors are finite, our method ensures the identifiability of the latent individual-specific factors, even when the transition processes are nonlinear. This establishes novel theoretical insights for learning state-transition processes with latent individual-specific factors.
>
> **Furthermore, motivated by the identifiability results, we have designed a practical generative framework, aligning with our theoretical assumptions, which is very different from VAE**. Specifically,
> - To satisfy the group determinacy assumption, which assumes there is a finite number of L, we utilize a quantized encoder where the embedding dictionary contains only a finite number of L.
> -  To satisfy the sequential dependency assumption, which assumes the generative process where $s_t$ is determined by $s_{t-1}$, $a_{t-1}$, L and noise, we model the noise with an MLP layer. Then, a conditional encoder, along with a noise layer, is designed to model the data generation process.
>
> [1] Zhang A, Lyle C, Sodhani S, et al. Invariant causal prediction for block mdps[C]//International Conference on Machine Learning. PMLR, 2020: 11214-11224.
>
> [2] Delgrange, Florent, Ann Nowe, and Guillermo A. Pérez. "Wasserstein Auto-encoded MDPs: Formal Verification of Efficiently Distilled RL Policies with Many-sided Guarantees." arXiv preprint arXiv:2303.12558 (2023).
>
> [3] Guo, Zhaohan Daniel, et al. "Bootstrap latent-predictive representations for multitask reinforcement learning." International Conference on Machine Learning. PMLR, 2020.
>
> [4] Hyvarinen A, Morioka H. Unsupervised feature extraction by time-contrastive learning and nonlinear ica[J]. Advances in neural information processing systems, 2016, 29.
>
> [5] Yao W, Sun Y, Ho A, et al. Learning temporally causal latent processes from general temporal data[J]. arXiv preprint arXiv:2110.05428, 2021.

---

> ### Author Response · Authors · 2023-11-19
> **Rebuttal by Authors [Part 2]**
>
> **Q2: Multi-Agent Reinforcement Learning methods and latent Markov Decision Process methods could also be applied to the setting described in this paper.**
>
> **A2:** It is hard to apply Multi-Agent Reinforcement Learning (MARL) and Latent Markov Decision Processes (MDPs) in our cases. The key points are summarized as follows:
>
> Regarding to Multi-Agent Reinforcement Learning
>
> **MARL typically involves multiple agents interacting in a shared environment, but in our work, each individual interacts with a unique environment. Therefore, MARL methods cannot be directly applied to our setting.** Specifically, in MARL, the actions of one agent can influence the outcomes of others, and agents jointly contribute to the dynamics of the environment. In contrast, in our work, each individual has a distinct environment, and there is no direct interaction between different individuals. The state transition processes are implicitly influenced by a latent individual-specific factor L.
>
> **If we tweak the MARL setting to allow different agents to have individualized environments, which could potentially be adapted to our setting, the agents would be trained independently.** In that case, the training would require more data, as all agents are trained separately in each environment, making it impossible to generalize across different environments in general. For example, the policy would need to be retrained when a new agent in a new individualized environment is introduced.
>
> Regarding to Latent Markov Decision Processes
>
>
>
> On the other hand, **in Latent MDP, latent variables (states) change with respect to time. However, in our work, we have a latent confounder L that influences each state in the transition process. All existing work for MDP with latent variables have not considered this specific case. Existing results cannot be directly applied. The details are as follows:
>
> - The state $ S_t $ usually refers to the specific situation of environment where the agent is present at timestep $ t $, and it only directly influences the subsequent state $ S_{t+1} $.
>
> - In contrast, $ L $  directly influences different states across all timesteps throughout the transition process.  $ L $ remains constant over time for each individual, while it may vary between individuals.  Mathematically, it is a latent confounder of all states.
>
> To the best our knowledge, we failed to find any works discuss such a latent confounder in the Markov decision process. Therefore existing work for MDP with latent variables cannot be directly applied here. These methods typically require that latent variables change over time [1-4]. The identifiability of having a latent confounder L of all states has not been explored in previous research.
>
>
> [1] Zhang, Xuezhou, et al. "Efficient reinforcement learning in block MDPs: A model-free representation learning approach." International Conference on Machine Learning. PMLR, 2022.
>
> [2] Guo, Zhaohan Daniel, et al. "Bootstrap latent-predictive representations for multitask reinforcement learning." International Conference on Machine Learning. PMLR, 2020.
>
> [3] Delgrange, Florent, Ann Nowe, and Guillermo A. Pérez. "Wasserstein Auto-encoded MDPs: Formal Verification of Efficiently Distilled RL Policies with Many-sided Guarantees." arXiv preprint arXiv:2303.12558 (2023).]
>
> [4] Zhang A, Lyle C, Sodhani S, et al. Invariant causal prediction for block mdps[C]//International Conference on Machine Learning. PMLR, 2020: 11214-11224.

---

> ### Author Response · Authors · 2023-11-19
> **Rebuttal by Authors [Part 3]**
>
> **Q3: What is the difference between individualized MDPs, partially observed MDPs and multi-agent MDPs.**
>
> **A3:**  We have made one-to-one comparisons as follows:
>
> - **POMDP  vs.  Individualized MDPs**
>   - POMDPs are similar to Latent MDPs. However, they do not consider our case where there is a latent confounder $L$ that influences each state in the transition process.
>
>   - **Incomplete Information vs. Latent Individual-Specific Factor.** POMDPs deal with situations where the agent has incomplete or uncertain information about the state of the environment. Such incomplete information is modeled as the hidden state space, which has a mapping to the observational space. Individualized MDPs, on the other hand, deal with fully observable states, but the observed state transition process is implicitly influenced by the latent individual-specific factor.
>   - **Time-Varying Influence on the Observation vs. Time-Invariant Influence on the Observation** In POMDPs, the latent states are time-varying, while in Individualized MDPs, the latent individual-specific factor is time-invariant for each individual.
>   -**Decision-making under Uncertainty vs. Decision-making for Individual Needs** POMDPs focus on scenarios where the agent must make decisions without complete information about the environment, while Individualized MDPs focus on adapting decisions to individual needs and preferences.
>
> - **MAMDPs vs. Individualized MDPs**
>   - **Agent Interaction vs. Individual Focus**: MAMDPs involve multiple agents whose actions affect each other, emphasizing the dynamics of all agents' interactions in one shared environment. In contrast, Individualized MDPs focus on optimizing decisions based on the unique characteristics of individual agents. The individuals share a common latent property but do not interact with each other.
>   - **Collective Optimization vs. Individualized Optimization**: MAMDPs often require strategies that balance collective outcomes for all agents, while Individualized MDPs aim to optimize outcomes for each individual, considering their specific context or requirements.
> (Please also refer **A2** for why method for MAMDPs hard to be applied to our cases).

---

> ### Author Response · Authors · 2023-11-19
> **Rebuttal by Authors [Part 4]**
>
> **Q4: The theoretical result requires unrealistic assumptions.**
>
> **A4:** We believe this is a misunderstanding. Our assumptions sometimes may not be satisfied but still realistic.
>
> To avoid confusion, we have added a detailed explanation of our assumptions in Appendix A.2. Specifically:
>
> > **Group determinacy** indicates that for a given task, the individual-specific factors are finite. This is a realistic assumption in numerous practical situations. For instance, in healthcare, consider two patients diagnosed with hypertension. Even if they receive the same treatment (or action $a_t$), they might experience different health outcomes (or state $s_t$). These differences can be attributed to latent factors like their medical history (or latent individual-specific factors $L$), which are essentially finite records of past medical events and treatments. In this case, we can group patients based on their history, creating categories such as those with no history of hypertension and those with a history of hypertension. Each group would demonstrate unique health outcomes and patterns, driven by their respective individual-specific factors $l_k$.
>
>
> > **Sequential dependency** is commonly seen in other related works, such as Temporal Causal Mechanisms in Block MDP [1]. It ensures that only previous information (such as state and action) influences the current state. In the case of individualized healthcare, this assumption implies that a patient's latent factors (such as medical history), current health state and current treatment decisions determine their next health state.
>
> > **Sample Sufficiency** indicates that sufficient data is needed to achieve identifiability, which should be a common assumption. For example, in nonlinear ICA using auxiliary variables [2], it assumes that there should be at least 2n+1 values for auxiliary variables to encourage variability; for disentanglement with minimal change [3], it assumes that there should be at least 2n+1 domain embeddings to ensure identifiability. Intuitively, without having data to provide us with related information about parameters, it is impossible to “determine” the value of these parameters.
>
> Note that even though the required assumptions may not always hold in practice, our method can still encourage identifiability in different cases, namely: (1) multiple independent latent confounders, (2) multiple independent continuous latent confounders, and (3) insufficient samples. A detailed discussion has been added in Section 4.3, and these propositions are empirically validated in the experiments in Section 5.1. Sepcifically,
>
> > In the first case, we can treat multiple latents as one discrete latent with cardinality $\prod^m_{i=1}c_i$. Then, each combination of the original multiple latents can be uniquely represented as one level in the combined latent variable. In the second case, the empirical results in Section 5 show that our proposed framework can also encourage the identifiability of the continuous latents. This is because we can divide the continuous latent confounder $L$ into small segments and represent each with a discrete value $l_k$. This discretization approach improves computational simplicity and stability. In the third case, even when the sample sufficiency assumption is weakly violated, our framework still encourages the identification of the latent confounder, as verified in the experiment in Section 5.
>
> [1] Zhang A, Lyle C, Sodhani S, et al. Invariant causal prediction for block mdps[C]//International Conference on Machine Learning. PMLR, 2020: 11214-11224.
>
> [2] Hyvarinen A, Sasaki H, Turner R. Nonlinear ICA using auxiliary variables and generalized contrastive learning[C]//The 22nd International Conference on Artificial Intelligence and Statistics. PMLR, 2019: 859-868.
>
> [3] Kong L, Xie S, Yao W, et al. Partial Identifiability for Domain Adaptation[J]. arXiv preprint arXiv:2306.06510, 2023.
>
> **Q5. The symbol L should depend on k in Equation (1).**
>
> **A5:** Thank you for pointing it out. We have updated Equation 1 and replaced $L$ with $l_k$.

---

> ### Author Response · Authors · 2023-11-19
> **Rebuttal by Authors [Part 5]**
>
> **Q6. Background information about VAE should be provided before applying (VQ-)VAE in Section 4.**
>
> **A6:** We have enriched Section 4 with additional background information on VAEs. To save space in the main paper, more detailed background on VAEs can be found in Appendix C. Specifically, the additions are cited as follows.
>
> > The traditional VAE encodes the input data into a continuous latent space using probabilistic encoders and then reconstructs the input from this space using decoders, which is not aligned with our problem setting.
>
> > Variational Autoencoders (VAEs) are a class of generative models in deep learning, adept at unsupervised learning of complex data distributions. Rooted in the framework of Bayesian inference, VAEs are designed to approximate probability density functions of input data. The architecture of a VAE consists of two primary components: an encoder $q_{\phi}(z|x)$ and a decoder $p_{\theta}(x|z)$. The encoder maps input data $x$ to a latent space, represented by a probability distribution, typically Gaussian, with parameters $\mu$ and $\sigma$ signifying the mean and standard deviation, respectively. The decoder reconstructs the input data from a sampled latent representation $z$.
>
>
> **Q7. The DDPG method is considered outdated. To show the significance and superiority.**
>
> **A7:** Thank you for highlighting this aspect. The major aim of **the experiment involving DDPG** was not to achieve state-of-the-art performance, but rather to demonstrate that incorporating individual-specific factors into policy learning enhances its effectiveness compared to approaches that do not include these factors. DDPG should be enough to validate this point. Empirically, with individual-specific factors, our method exhibits faster generalization in policy learning and achieves better performance.
> Note that our framework is generally and can be integrated with various reinforcement learning (RL) algorithms. We expect that incorporating the latest advancements in RL will further improve the performance of our method.
> Additionally, we have revised Section 4.2 to emphasize the design of individualized policies using DDPG as an example.
>
> > In general, the conventional policy input has been extended to include the latent individual-specific factor to each individual, and the training objective has been adjusted accordingly to prioritize the optimization of policies tailored to the individual.
>
>
> **Q8. The font size in Figure 3 is too small to be readable.**
>
> **A8:** Thank you for pointing that out. We have increased the font size in Figure 3 to make it easier to read.
>
>
>
> **Q9: What exactly does the identifiability property of the individualized latent factor in Theorem 1 mean.**.
>
> **A9:** An intuitive explanation of identifiability is that a latent individual-specific factor is said to be identifiable if no other latent factor (of equal or lesser complexity) can explain (or generate) the distribution of observed variables.
>
> In our cases, L is considered identifiable if the joint distribution derived from a learned generative model matches the true model everywhere, and such observational equivalence can always lead to the identifiability of the latent variables. This refers to the ability to uniquely and accurately determine latent factors from observed data. We have provided the definition of identifiability in Appendix A.3.1.
>
> **Q10: The optimal policy appears to be non-stationary in individualized MDPs. How can this difficulty be addressed.**
>
> **A10:** We guess the non-stationarity you mentioned refers to the policy varying across individuals, considering that changes in the latent factors lead to changes in the transition dynamics, whereas the individualized policy is stationary for each individual.
>
> To address this issue, we have included the latent individual-specific factor as the policy input and adjusted the training objective accordingly to prioritize the optimization of individual policies. When a new individual is encountered, the agent simultaneously optimizes its policy and collects new data to estimate the latent factors. The details have been added in Section 4.2 and are as follows:
>
> > After optimizing the individualized policy, the agent uses $\pi^*_k$ as a warm start, which is an initial starting point that helps speed up training, and continues training on the new individual. To achieve zero-shot transfer, we simultaneously optimize the individualized policy while interacting with the environment. Thus, adaptation involves a dual process: it fine-tunes the policy based on interactions with the new individual, and it actively collects data from these interactions. This data collection is critical for estimating the latent individual-specific factor of the new individual. During this phase, the policy is continuously adjusted and improved, making it better suited to each new individual.

---

> ### Author Response · Authors · 2023-11-22
> **Could you please let us know whether our responses and updated submission properly addressed your concern?**
>
> Dear Reviewer 2bUe,
>
> Thank you once again for your time dedicated to reviewing our paper and for your insightful suggestions. We hope your comments have been properly addressed, and look forward to receiving your feedback at your earliest convenience. We are more than happy to respond to it.
>
> Sincerely,
>
> Authors

---

> > ### Comment · Reviewer_2bUe · 2023-11-22
> >
> > I really appreciate the detailed response. I have read the updated paper and find that the writing has improved but is still hard to follow for general readers. Here are some comments:
> >
> > - My concerns regarding the differences with POMDP, latent-MDP, and MARL are well-addressed.
> >
> > - It is not advisable to introduce undefined terminologies (e.g., identifiability definition A.3, the group determinacy assumption) in the overview paragraph in Section 4, as they just confuse readers.
> >
> > - A typo exists on Line 3 of Page 7: "For instance in hypertension diagnosed. Even if" should be "For instance in diagnosed hypertension, even if".
> >
> > - The reviewer understands that this paper aims to use population data to estimate $q$ individual MDPs. There are two practical questions: how is the value $q$ defined in advance? What happens if the group determinacy assumption does not exist? The latter case corresponds to the fact that the proof of Theorem 1 relies on the Lemma of (Vandermeulen & Scott, 2015), which resloves the problem of identifying mixture models from grouped samples. In general, the latent confounders are mixed rather than independent.

---

> > > ### Author Response · Authors · 2023-11-23
> > > **Responses by Authors to Follow-Up Questions**
> > >
> > > Dear Reviewer 2bUe,
> > >
> > > Thanks once again for your recognition of our revision! We are more than happy to address your concerns in details below.
> > >
> > > **Follow-Up Q1: Introduce undefined terminologies (e.g., identifiability definition A.3, the group determinacy assumption) in the overview paragraph**.
> > >
> > > **Follow-Up A1:** Thank you for your insightful suggestion. We have relocated the context related to the theorem to Section 4 (a new theorem-related section), before introducing our framework in Section 5. We hope this will enhance the readability of our paper.
> > >
> > > **Follow-Up Q2: A typo exists on Line 3 of Page 7**.
> > >
> > > **Follow-Up A2:** We apologize for the typo and we have corrected it in the revised version.
> > >
> > > **Follow-Up Q3: How is the value $q$ defined in advance?**
> > >
> > > **Follow-Up A3:** Thank you for your interesting question! In our paper, we consider a scenario where a population of $K$ unique individuals is divided into $q$ groups. In that case, the value of $q$ is pre-determined (see the Data Generation Process in Preliminaries).
> > >
> > > > Consider a population with $K$ unique individuals that can be divided into $q$ groups, where the exact group membership is unknown.
> > >
> > > In practice, the definition of the $q$ value depends on how the population is grouped, which may vary according to specific downstream tasks. For instance, in a patient population, grouping might be based on disease severity when focusing on treatment efficacy. Conversely, for studies on genetic influences, patients may be categorized by family history or genetic markers.
> > >
> > > In our experiment, we set different $q$ values for different environment settings. Specifically, we set $q$ to 4 for the synthetic experiment in Case 1, and to 5 for Pendulum control. In Persuasion For Good Corpus, all participants underwent personality assessments, and we are lucky to have labeled 35-dimensional individualized personalities for each persuadee.
> > >
> > > **Follow-Up Q4: What happens if the group determinacy assumption does not exist?**
> > >
> > > **Follow-Up A4:** Thank you for your valuable insight! The group determinacy assumption implies the existence of a latent individual-specific factor $L$. We constrained the formation of $L$ to proceed with the proof, which might not align with practical situations. We acknowledge this as a limitation in our work and have discussed it in our conclusion. Extending our identifiability theories and framework to encompass broader properties is our future direction. Specifically,
> > >
> > > > The main limitations of this work lie in our two main assumptions: (1) there is no instantaneous causal influence within $\vec{s}\_t$, and (2) the latent confounder is discrete. Instantaneous relationships, if present, may distort the identifiability of the latent state-transition process, although their impact can be moderated by adjusting the temporal resolution of the data. Although our theoretical framework does not extend to scenarios with evolving continuous latents, our empirical results suggest potential for adapting our approach to a broader range of scenarios. Extending our identifiability theories and framework to account for such properties is our future direction.

---

### Official Review · Reviewer_DHA9 · 2023-11-09

**Soundness:** 2 fair
**Presentation:** 2 fair
**Contribution:** 2 fair
**Rating:** 6
**Confidence:** 3

**Summary:**

The paper examines the application of reinforcement learning (RL) in personalized settings such as healthcare and education, where individual differences significantly impact the outcomes of RL agents. It addresses the challenge of identifying individual-specific factors that influence state transitions, which are often hidden (latent). The authors propose a novel method for learning these unique state-transition processes from observed interactions, with the significant advantage of theoretical support for its effectiveness. This approach is touted as the first to offer such guarantees. The paper validates the method with both synthetic and real-world data, showing its potential in crafting tailored RL policies. This contribution is particularly relevant for developing RL applications that must adapt to individual user characteristics, a common scenario in personalized services.

**Strengths:**

The paper addresses a critical research question in personalized reinforcement learning (RL) by focusing on the identification of latent state-transition processes. This is a significant contribution to the field, as it tackles the complexities of individual differences that are pivotal in RL outcomes but are often not directly observable.

**Weaknesses:**

The paper presents a methodological approach to personalized reinforcement learning, yet there are several areas where clarity and depth are lacking. Specifically, Section 4 would benefit from a citation on methods used to enhance user experience on multimedia platforms, which is currently missing. In Section 3, the description of the reward function \( R \) is incomplete; it is essential to clarify its form and relationship with other variables, which is not currently addressed.

Remark 1 appears verbose and could be made more succinct to help readers grasp the core concept more rapidly. Additionally, Section 4's text is overly descriptive and lacks the necessary equations or algorithmic details to clearly understand the proposed processes, such as noise estimation and its application within the model architecture.

The paper's novelty is also a point of concern; the work appears incremental and relies heavily on established frameworks and methods without offering new insights or innovative approaches. The rationale behind the use of a noise module following discrete encoding is not clear, and the similarity to continuous VAE processes needs to be justified.

Overall, the manuscript would benefit from a thorough polish to enhance readability and flow. The current state may present difficulties for readers in following the progression of ideas and fully understanding the proposed methods.

**Questions:**

Please refer to weaknesses.

---

> ### Author Response · Authors · 2023-11-19
> **Rebuttal by Authors [Part 1]**
>
> **Q1: Section 4 would benefit from a citation on methods used to enhance user experience on multimedia platforms, which is currently missing.**
>
> **A1:** Thank you for pointing this out. We guess you were referring to the two missing citations [1-2] in Section 2, which relate to user experience on multimedia platforms. We have included them in the revised version.
>
> [1] Hoiles W, Krishnamurthy V, Pattanayak K. Rationally inattentive inverse reinforcement learning explains youtube commenting behavior[J]. The Journal of Machine Learning Research, 2020, 21(1): 6879-6917.
>
> [2] Cai Q, Zhan R, Zhang C, et al. Constrained reinforcement learning for short video recommendation[J]. arXiv preprint arXiv:2205.13248, 2022.
>
>
>
> **Q2: The description of the reward function is incomplete.**
>
> **A2:** Thanks for pointing this out. We have revised the definition of Individualized Markov Decision Processes in Section 3 and defined a general form for the reward function as follows:
> > $r_t=R(\vec{s}\_{t-1},\vec{a}\_{t-1},\vec{s}\_t)\in\mathbb{R}$ denotes the immediate reward.
>
> Note that the specific reward function depends on the downstream control task, which is hard-coded in the environments. In pendulum control, the reward function is defined as $r_t=-(\theta_t^2 + 0.1 * \omega_t^2 + 0.001 * a_t^2)$. We have provided it in Appendix D.2 in the revised version.
>
> The other synthetic dataset and Persuasion For Good Corpus dataset in our paper are only used to validate whether our method can successfully infer the latent individual-specific factor, which is not a control task. There is no reward function.
>
>
>
> **Q3: The paper presents a methodological approach to personalized reinforcement learning, yet there are several areas where clarity and depth are lacking.**
>
> **A3:** In **A1** and **A2**, we believe we have addressed the specific issues you raised regarding this concern, **specifically the missing citations and the incomplete reward function**. However, we are uncertain if these issues have been completely resolved to your concern. Please kindly inform us if there are any remaining concerns.
>
>
>
>
> **Q4: Remark 1 appears verbose and could be made more succinct to help readers grasp the core concept more rapidly.**
>
> **A4:** Thank you for your feedback. To make the paper more concrete and easier to understand, we have removed Remark 1 and merged it with the motivation example in Section 3. We also provide an illustration Figure 1(b) to explain our motivation.
>
> The specific changes are as follows:
> > This formulation is intuitive in many real-world applications. For example, in the context of hypertension diagnosis as shown in Figure 1(b). The patient's health status at each time can be represented as the state $\vec{s}\_t$, the individualized treatment can be viewed as the action $\vec{a}\_t$, and the blood pressure level is the reward $r_t$. The latent confounder $l$ represents the unobserved genetic predispositions that are unique to each patient and have a significant impact on their long-term health, such as gene variations. In this case, the health status $\vec{s}\_t$ at time $t$ is determined by the previous status $\vec{s}\_{t-1}$, previous treatment $\vec{a}\_{t-1}$, together with the intrinsic gene variations. If we could observe these predispositions, it would greatly help physicians to predict how patients will respond to specific treatments, enabling personalized care. In most cases, however, these individual-specific factors remain latent and difficult to diagnose. Therefore, the theoretical identification of such traits and the development of an estimation framework to extract them from observed data is of great practical value.

---

> ### Author Response · Authors · 2023-11-19
> **Rebuttal by Authors [Part 2]**
>
> **Q5: Section 4 lacks the necessary equations or algorithmic details, such as noise estimation and its application within the model architecture.**
>
> **A5:** Thank you for your feedback. We have thoroughly revised Section 4 for better clarity and comprehension.
>
> For the noise estimation part, we revised as follows:
> > To account for the stochastic nature of the individualized transition processes, a noise estimator is introduced with $\hat{l}^k$ as input and $\hat{\nu}_k$ as output. This integration attempts to fine-tune the reconstruction of the next states $\hat{\vec{s}}\_t$ and decrease the reconstruction error with the actual next state $\vec{s}\_t$.
>
> For its application in our model, we revised as follows:
> > By introducing noise, we model the stochasticity in our defined transition process. Empirically, this noise is modeled using an MLP (Multilayer Perceptron) layer. Then, the output of the noise layer is used as the input of the conditional decoder,
> along with the conditions $(s_{t-1}, a_{t-1})$, and the latent individual-specific factor $L$. The conditional decoder outputs $s_t$ and is designed to model the data generation process.
>
> Additionally, to further improve the clarity of Section 4, we made the following changes in the revision.
> - We have redrawn Figure 2 to better illustrate the structure of the framework;
> - We have **revised the overview paragraph** to explain the methodology's design and emphasize the connection between the proposed framework and the identifiability theorem. Specifically, the overview has been revised as follows:
>
> > Our proposed framework is carefully customized to meet the requirements of the identifiability theorem. According to the identifiability definition A.3, the latent confounders are identifiable if observational equivalence can always lead to the identifiability of the latent variables. This motivates our work to use a generative model and to realize confounder estimation by fitting the learned distribution to the true observed distribution. The traditional VAE encodes the input data into a continuous latent space using probabilistic encoders, and then reconstructs the input from this space using decoders, which is not suitable for our problem setting. Specifically, given the offline trajectory of the population, we customize the VAE framework and develop an individual-specific factor estimation framework to recover these factors.
>
> - We have **included essential equations and algorithmic insights in each module.** Specifically,
> -  In the quantized encoder module, we have added the motivation for introducing the quantization layer, along with update functions as follows:
>
> > The group determinacy assumption implies the existence of the latent individual-specific factor $L$. Since $L$ denotes the time-invariant latent individual-specific factors that influence each state in the transition process, we initially use an encoder to capture the continuous representation $z^k$ from $\vec{s}^k_{0:T}$, and employ a quantization layer to delineate unique and distinct groups as $\hat{l}^k$.
>
> > The corresponding update functions are $o^k_t=\text{Conv1D}(\vec{s}^k_{t:t+H}),$ $h^k_t,c^k_t=\text{LSTM}(h^k_{t-1},c^k_{t-1},\vec{s}^k_t;\theta),$ where $o^k_t$, $h^k_t$ and $c^k_t$ are the feature map, hidden state and cell state at time $t$, and $\theta$ denotes all the parameters of the LSTM.
>
> **In the noise estimation module, we have provided the motivation for including this module and explained how it applies within our framework.**
> > To account for the stochastic nature of the individualized transition processes, a noise estimator is introduced with $\hat{l}^k$ as input and $\hat{\nu}_k$ as output.
>
> > The motivations for introducing a noise layer are 1) to mimic the probabilistic property of the individualized state transition in Equation 1, and 2) to compensate for the loss of stochasticity due to the deterministic nature of the quantization operation.
>
> > Our posterior $ q(\hat{l}\_k=\kappa|\vec{s}^k_{0:T}) = \\{1, \text{for } \kappa=\arg\min\_j \lVert z^k-e\_j\lVert\_2; 0\text{ otherwise.}\\}$,  is a categorical distribution and becomes deterministic after the quantization operation.By introducing noise into the decoder, we introduce variability that 1) satisfies our individualized transition processes and 2) enables the model to better vary data, thus improving the model's ability to generalize and increasing robustness in different scenarios.
>
> - Furthermore, we have added a new paragraph and introduced the training objective to explain how to optimize the entire framework.
>
> > During the training process, the parameters are optimized according to the extended ELBO objective $\mathcal{L}\_{\rm{ELBO}}$.
> $\mathcal{L}\_{\rm{ELBO}} = \mathcal{L}\_{\rm{Recon}} + \beta\mathcal{L}\_{\text{Quant}} + \alpha\mathcal{L}\_{\text{Commit}}$, where $\alpha$ and $\beta$ are weights for the corresponding loss components.

---

> ### Author Response · Authors · 2023-11-19
> **Rebuttal by Authors [Part 3]**
>
> **Q6: The work appears incremental and relies heavily on established frameworks and methods, therefore the paper's novelty is a point of concern.**
>
> **A6:** We apologize for any confusion. In this paper, we answer two key questions: (1) when can the latent individual-specific factors be sufficiently identified, and (2) how can we learn them effectively? We believe that the answers to these questions are not trivial. Providing clarity on these matters is important and has the potential for many applications.
>
> **Here, we would like to provide details about the differences between our method and others, and highlight our novelty.**
>
> To the best of our knowledge, none of the existing works consider time-invariant latent individual-specific factors that influence each state in the transition process. We are the first to model such processes and provide an identifiability result.
>
> In this work, we aim to bridge this gap. We seek to answer the important question of which types of transition processes can have a theoretical identifiability guarantee. We demonstrate that when the latent individual-specific factors are finite, our method ensures the identifiability of the entire latent state-transition process, even when the transition processes are nonlinear. This establishes novel theoretical insights for learning state-transition processes with the latent individual-specific factors.
>
> Furthermore, motivated by the identifiability results, we have designed a practical generative framework that aligns with our theoretical assumptions, **which is very different from a VAE**. Specifically,
>
> 1. To satisfy the group determinacy assumption, which assumes there is a finite number of L, we utilize a quantized encoder where the embedding dictionary contains only a finite number of L.
>
> 2. To satisfy the sequential dependency assumption, which assumes the generative process where $ s_t $ is determined by $ s_{t-1} $, $ a_{t-1} $, L, and noise, we model the noise with an MLP layer. Then, a conditional encoder, along with a noise layer, is designed to model the data generation process.
>
> **To enable readers to quickly grasp the essence of our contribution, we have revised the Introduction as follows:**
> > We propose the Individualized Markov Decision Processes (iMDPs) framework, a novel approach that integrates individual-specific factors into decision-making processes. We model these factors as latent confounders. We allow them to influence each state in the decision process and to vary across different individuals. This framework has many real-world applications.
>
> > Our framework has theoretical guarantees. We show that when individual-specific factors are finite, our method ensures the identifiability of whole latent state-transition processes, even when the transition processes are nonlinear. This establishes novel theoretical insights for learning state-transition processes with latent individual-specific factors. Additionally, for scenarios with non-finite individual-specific factors, we show that categorizing them into finite groups has little impact on the empirical results.
> To the best of our knowledge, this is the first work to provide a theoretical guarantee for the identification of latent individual-specific factors from observed data.
>
> > We propose a practical generative-based method that can effectively infer latent individual-specific factors. Empirical results on both synthetic and real-world datasets demonstrate the method's effectiveness not only in inferring these factors but also in learning individualized policies.
>
>
> **Q7: The manuscript would benefit from a thorough polish to enhance readability and flow.**
>
> **A7:** Thank you for your constructive comments. We have carefully revised the manuscript according to your suggestions to enhance its readability and logical flow. A brief summary of change is as follows
> - In Section 1, we emphasized our contributions.
> - In Section 3, we reaffirmed the motivation and strengthened the connection of our model to real-world applications.
> - In Section 4, we thoroughly revised the framework, highlighted its connection with theoretical guarantees, and improved the presentation.
>
> We have tried our best to address your concern. Please kindly let us know if you have any further issues.

---

> ### Comment · Reviewer_DHA9 · 2023-11-21
>
> Thank you for your comprehensive feedback. I appreciate that several of my concerns have been addressed, prompting me to increase my rating from 3 to 5. Nonetheless, there are an additional aspect I hope the author will consider. Addressing these effectively could lead to a further increase in my rating:
>
> 1. Providing an algorithm in pseudocode that integrates all equations and loss computations in the method section would significantly improve understanding of the overall paradigm.
> 2. Could you clarify the distinction between a VQ-VAE with an added continuous noise estimator and a VAE that directly generates continuous embedding vectors?

---

> ### Author Response · Authors · 2023-11-21
> **Responses by Authors to Follow-Up Questions**
>
> Dear Reviewer DHA9,
>
> Thank you for acknowledging our revisions! We are more than happy to address your concerns in detail below.
>
> **Follow-Up Q1: Provide an algorithm in pseudocode that integrates all equations and loss computations.**
>
> **Follow-Up A1:** Thanks for your valuable advice! We have revised the pseudocode in Appendix C. The revised version provides a more detailed description for each module and includes all the equations and loss computation mentioned in the main paper.
>
> **Follow-Up Q2: The distinction between a VQ-VAE with an added continuous noise estimator and a VAE that directly generates continuous embedding vectors.**
>
> **Follow-Up A2:** We apologize for the confusion. In our work, **the noise estimator's output serves as an additional input to the conditional decoder, rather than being directly added to the estimated individual-specific factor $\hat{l}$**. Consequently, $\hat{l}$ remains a discrete representation in this context, aligning with our group determinacy assumption and state-transition process.
>
> It is worth noting that we consider that different individuals may have different distributions of the noise. Therefore, we let the noise depend on $\hat{l}$ and use $\hat{l}$ for noise generation.
>
> Additionally, to improve clarity, **we have revised the description in Section 4 and the comparison of our framework with traditional VAEs in Appendix E**. Specifically:
>
> > By adding noise to the decoder, we introduce variability that 1) satisfies our individualized transition processes, and 2) allows the model to better vary the data. The empirical results of the ablation study validate the model's enhanced ability to improve generalization.
>
> > The traditional VAE encodes input data into a continuous latent space using probabilistic encoders, and then reconstructs the input from this space using decoders. Our proposed framework differs in three main aspects:
> > 1) Our framework uses a quantization layer to discretize the continuous latent representations. This mapping of continuous latent representations to an embedding dictionary is well suited to the group determinacy requirement.
> > 2) We estimate noise as an input to the decoder to introduce variability that satisfies our individualized transition processes and allows the model to better vary the data.
> > 3) Our decoder reconstructs individualized transition processes to simulate the data generation process, incorporating additional conditions, estimated latent individual-specific factor, and noise.

---

> > ### Comment · Reviewer_DHA9 · 2023-11-22
> >
> > Thanks for your feedback! I think after rebuttal my concerns are all addressed. Therefore, I will raise my score to 6.

---

> > > ### Author Response · Authors · 2023-11-22
> > > **Thanks**
> > >
> > > Dear Reviewer DHA9,
> > >
> > > Thank you very much for your efforts. Your comment is really helpful in improving our paper! All changes related to your comments will be carefully addressed in our final version.
> > >
> > > Warm regards,
> > >
> > > Authors

---

### Author Response · Authors · 2023-11-20
**Thanks to All Reviewers. Are There Any Other Concerns?**

Dear Reviewer DHA9, Reviewer 2bUe, Reviewer rs8u, Reviewer y8Yq, and Reviewer Mxvj,

**Thank you very much for reviewing our paper. We greatly appreciate all your efforts in helping us.**

We have tried our best to address all your concerns and have revised our paper accordingly. Major revisions are as follows.

- Emphasizing the contributions in Section 1 to highlight the novelty.

- Reorganizing the preliminaries in Section 3 to clarify the problem statement.

- Revising the methodology in Sections 4.1 and 4.2 to improve clarity and emphasize the design motivation.

- Adding explanations on assumptions in Section 4.3 to enhance understanding.

- Revising the experimental settings in Section 5 to enhance the paper's reproducibility.

We are grateful for the reviewers' recognition of the *motivation* and *well-crafted* nature of our paper. Please note that this is the first work to model and illustrate identifiability results of state-transition processes involving latent confounders that directly influence all states, which has the potential for many applications.

We would greatly appreciate it if the reviewers could reconsider the score based on our revised version.

Please let us know if there are any further concerns, and we are more than happy to address them.

Sincerely,

Authors

---

### Meta-Review · Area_Chair_135S · 2023-12-09

**Metareview:**

The paper introduces an approach for individualized reinforcement learning through the learning of latent state-transition processes using a Variational Autoencoder (VAE). The reviewers and authors have provided comprehensive discussion, highlighting or addressing several concerns. However, the reviewers were not excited about this paper and thought some of the novelty were over-claimed. Therefore, it leads to the recommendation for rejecting the paper.

Overall, the reviewers express concerns about clarity, methodology, experimental design, and the extent of the contribution. There are questions about assumptions, motivations, and the novelty of the proposed approach. While some strengths, such as clear writing and the provision of code, are acknowledged, the weaknesses and uncertainties raised by the reviewers suggest the  rejection of this paper.

**Justification For Why Not Higher Score:**

The reviewers were not excited about this paper and thought some of the novelty were over-claimed.

**Justification For Why Not Lower Score:**

N/A

---

### Decision · Program_Chairs · 2024-01-16

Reject